# AttentionPredictor: Temporal Patterns Matter for KV Cache Compression

**Qingyue Yang**[1][*], **Jie Wang**[1][†], **Xing Li**[2][‡], **Zhihai Wang**[1], **Chen Chen**[2], **Lei Chen**[2],
**Xianzhi Yu**[2], **Wulong Liu**[2], **Jianye Hao**[2, 3], **Mingxuan Yuan**[2], **Bin Li**[1]

[1]MoE Key Laboratory of Brain-inspired Intelligent Perception and Cognition,
University of Science and Technology of China
[2]Huawei Noah's Ark Lab
[3]College of Intelligence and Computing, Tianjin University
{yangqingyue, zhwangx}@mail.ustc.edu.cn,
jiewangx@ustc.edu.cn, li.xing2@huawei.com

## Abstract

With the development of large language models (LLMs), efficient inference through Key-Value (KV) cache compression has attracted considerable attention, especially for long-context generation. To compress the KV cache, recent methods identify critical KV tokens through static modeling of attention scores. However, these methods often struggle to accurately determine critical tokens as they neglect the *temporal patterns* in attention scores, resulting in a noticeable degradation in LLM performance. To address this challenge, we propose **AttentionPredictor**, which is the **first learning-based method to directly predict attention patterns for KV cache compression and critical token identification.** Specifically, Attention-Predictor learns a lightweight, unified convolution model to dynamically capture spatiotemporal patterns and predict the next-token attention scores. An appealing feature of AttentionPredictor is that it accurately predicts the attention score and shares the unified prediction model, which consumes negligible memory, among all transformer layers. Moreover, we propose a cross-token critical cache prefetching framework that hides the token estimation time overhead to accelerate the decoding stage. By retaining most of the attention information, AttentionPredictor achieves **13×** KV cache compression and **5.6×** speedup in a cache offloading scenario with comparable LLM performance, significantly outperforming the state-of-the-arts. The code is available at https://github.com/MIRALab-USTC/LLM-AttentionPredictor.

## 1 Introduction

Large language models (LLM) like OpenAI o1 [1] have shown impressive scalability and effectiveness in tackling complex tasks through long chain-of-thought (CoT) reasoning and multi-turn conversations [2–5]. However, these long-context tasks require LLMs to handle extremely lengthy contexts, presenting computational and memory challenges for LLMs [6]. Specifically, the key-value cache (KV cache), which holds the attention keys and values during generation to prevent re-computations, consumes huge GPU memory [7]. For example, for a model with 7 billion parameters, the parameters consume only 14 GB of memory whereas the KV cache requires around 72 GB with the 128K prompt length [8]. As the decoding latency and memory footprint scale with the KV cache, it is important to compress the KV cache as the prompt expands.

---

[*]This work was done when Qingyue Yang was an intern at Huawei.

[‡]Project lead.

[†]Corresponding author. Email: jiewangx@ustc.edu.cn.

39th Conference on Neural Information Processing Systems (NeurIPS 2025).

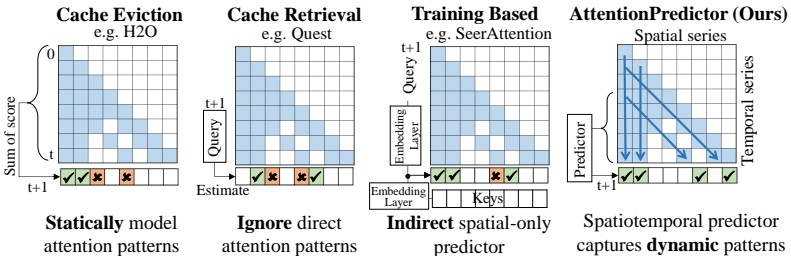

Figure 1: A comparison of H2O, Quest, SeerAttention, and AttentionPredictor for identifying critical tokens in the next step with history attention score. Our learning-based spatiotemporal predictor captures the dynamic attention patterns and accurately predicts next-step attention scores.

Many attention sparsity-based methods have been proposed to compress the KV cache in the sequence dimension. Previous works have shown that a small portion of tokens dominate the attention distribution, substantially determining token generation precision [9–11]. Therefore, we can dramatically reduce the cache size by computing attention sparsely with only the critical tokens, while maintaining LLM performance. Cache eviction methods [11–13] use heuristic ranking with attention scores to identify critical keys and evict the less relevant ones. However, heuristic scoring methods can only model **fixed patterns** and struggle to identify critical tokens accurately, causing these methods to suffer from LLM performance degradation. Recently, SeerAttention [14] and Attention-Gate [15] use learnable modules to model **dynamic patterns** and retrieve critical tokens. However, they only represent keys or hidden states, rather than directly modeling the attention score distribution. As a result, they often exhibit limited accuracy after compressing KV cache (Table 8 for details). This motivates the need for a more effective solution that directly targets the prediction of attention scores for enhanced KV cache compression.

In our paper, we thus explore the importance of **temporal patterns** in attention scores for identifying critical KV cache. Previous research indicates that attention scores exhibit repetitive patterns intrinsic to LLMs [10, 16–18]. We further observe that attention patterns have temporal characteristics, such as re-access, sequential, and seasonal, as illustrated in Figure 2. Our theoretical analysis in Section 3.3 shows these patterns originate from intrinsic model properties such as query similarity and position encoding. These patterns are both stable and predictable across time, forming a two-dimensional (2D) temporal map with local continuity and translation-invariant characteristics.

KV cache compression with critical token identification can be formulated to attention score prediction which is a 2D time series prediction problem. Inspired by the theoretical understanding of attention mechanisms, we introduce **AttentionPredictor**, a time-series prediction approach to predict attention scores for critical token identification, as shown in Figure 1. Since temporal attention patterns are dynamic and difficult to capture with existing heuristic methods, we learn lightweight but accurate prediction models to address this challenge. Specifically, we design a convolutional module to capture 2D patterns in the input attention scores, allowing a single predictor to be shared across all transformer layers. These choices shrink our unified prediction model to roughly **one millionth the size of the LLM** (LLaMA-3.1-8B), resulting in negligible memory consumption. In contrast, SeerAttention's per-layer prediction model amounts to 101MB in total, whereas ours is only 21KB—just 0.02% of its size. Additionally, we employ distribution error calibration by periodically computing dense attention, and apply block-wise attention compression to further improve the efficiency of prediction. By accurately identifying critical tokens, AttentionPredictor retains most of the attention information after KV cache compression. To further accelerate LLM decoding, we apply AttentionPredictor to our proposed KV cache management framework, a **cross-token KV cache prefetching** system. In contrast to the existing cross-layer approach [19], our cross-token method can capitalize on longer transfer times, and adapt to more sophisticated and accurate methods for identifying critical tokens.

Our contributions: 1) Based on our observation of the temporal patterns in the attention score, we propose AttentionPredictor. To the best of our knowledge, it is **the first learning-based method to directly predict attention patterns for KV cache compression and attention sparsity.** 2) We propose the first cross-token prefetching framework, which effectively mitigates prediction delays and KV cache transfer latency in the LLM decoding stage. 3) Experiments demonstrate that our approach achieves **13×** KV cache compression with comparable LLM performance and **5.6×** speedup to full cache offloading.

## 2 Related works

### 2.1 Efficient LLM Inference

Several efforts have optimized pre-trained model inference efficiency from different perspectives. Speculative decoding methods [20–24] use smaller draft models to predict multiple future tokens for parallel validation by a target LLM. In contrast, our approach instead predicts the next token's attention scores to estimate KV cache compression. Inference systems [25–29] accelerate the prefilling stage with techniques like prefix cache. AttentionPredictor is compatible with them, for it is applied to the decoding stage. At the same time, our block-based sparsification can also be applied to paged attention [25] and other computational accelerations.

### 2.2 KV Cache Compression

Many methods compress the KV cache by leveraging the finding that attention scores are sparse, which allows for estimated sparse computation on high-score positions.

**Cache eviction.** These methods use heuristic approaches to identify key-value pairs of high importance and evict the less relevant ones. StreamingLLM [12] observes that the earliest tokens have high attention scores during inference, so it only retains the initial and the recent few tokens. H2O [11] accumulates all historical attention scores as an estimation, but suffers from the accumulation error of scores from the first few tokens being too frequent. SnapKV [13] accumulates attention scores within a recent window as an estimate and performs one-time filtering during the prefill stage. MInference [16] and FlexPrefill [30] inductively define several attention patterns and determine the compression strategy for each head to accelerate the prefilling stage. All these methods are heuristic-based and statistically model attention patterns. They struggle to capture the dynamic temporal patterns within attention scores accurately.

**Cache retrieval.** These methods, such as Quest [31], InfLLM [32], and PQCache [33], retrieve critical tokens by estimating attention weights over compressed key representations. However, such compression can degrade fidelity and introduce retrieval errors. In particular, Quest is sensitive to the page size, and the accuracy significantly drops with large page sizes and small budgets, as shown in Figure 8. Instead, our method employs a lightweight temporal predictor to estimate attention weights directly, yielding higher retrieval accuracy. Other methods involving KV cache quantization [18, 34–36] and KV cache budget allocation [37, 8, 38, 39] are orthogonal to our token scoring approach and can be combined with our AttentionPredictor.

**Training-based Approaches.** Recently, a number of training-based cache compression approaches have been proposed. MoBA [40] extends the cache retrieval method by integrating sparse attention during training. Similarly, SeerAttention [14] and NSA [41] both train linear models to encode and retrieve key blocks more accurately. However, these approaches typically require training a separate model for each layer, which limits their scalability and generalization. In contrast, our framework employs a single plug-and-play module that can be applied uniformly to all layers and heads within the same LLM. Our model is only **0.02%** the size of SeerAttention. Moreover, whereas NSA relies on the fine-tuning of LLMs to achieve optimal results, our method achieves comparable improvements without any additional fine-tuning, making it both more efficient and broadly applicable.

**KV cache cross-layer prefetching.** These methods hide part of the cache transfer time based on offloading the cache to the CPU. However, as the sequence length increases, the transfer time is too long to be hidden. InfiniGen [19] combines cache retrieval and prefetching by approximating the attention score for the next layer to load the critical cache. However, the estimation time increases significantly as the sequence grows, and the inference time for a single layer is insufficient to cover this. In contrast, our cross-token prefetching framework can hide longer estimation and transfer time within the per-token inference time.

## 3 Method

In this section, we introduce AttentionPredictor, **the first learning-based method to directly predict attention patterns for KV cache compression and attention sparsity.**, along with the cross-token prefetch framework for improved cache management. We begin with the problem formulation for attention prediction in Section 3.2, followed by the observation and theoretical analysis of temporal

attention patterns in Section 3.3. Then we introduce our novel AttentionPredictor in Section 3.4. Finally, Section 3.5 presents a cross-token prefetch framework that efficiently hides both evaluation and cache loading latencies.

## 3.1 Preliminary

In the language model decoding stage, we denote $\mathbf{Q}_t \in \mathbb{R}^{1 \times d}$, $\mathbf{K} \in \mathbb{R}^{t \times d}$ as the query tensor and key tensor used for generate token $t$, respectively. Specifically, we denote $\mathbf{K}_i \in \mathbb{R}^{1 \times d}$, where $i \in \{1, 2, \ldots, t\}$, as the key tensor for token $i$, and $\mathbf{K} = \mathbf{K}_{1:t}$ as the complete key tensor. The attention score at step $t$ is calculated as: $A_t = \text{Softmax}\left(\frac{1}{\sqrt{d}}\mathbf{Q}_t\mathbf{K}^\top\right) = [a_{t,1}, a_{t,2}, \ldots, a_{t,t}] \in \mathbb{R}^{1 \times t}$.

The sparsity-based KV cache compression seeks to find a subset of keys with budget $B$ that preserves the most important attention values. Specifically, the subset of key indices is $S = \{s_1, s_2, ..., s_B\}$, where each $s_j$ is an index within the total $t$ tokens. We define the **attention recovery rate** as:

$$R_{rec} = \frac{\sum_{j=1}^{B} a_{t,s_j}}{||A_t||_1}, \tag{1}$$

which reflects the amount of information preserved after compression. A higher recovery rate $R_{rec}$ indicates less information loss caused by KV cache compression. Therefore, the goal of KV cache compression can be formulated as finding the index set $S$ that maximizes $R_{rec}$.

## 3.2 Problem Formulation

Accurate next-step attention score modeling is critical to maintain high attention recovery rate with limited KV tokens and improve final LLM generation performance with KV cache compression. Therefore, **KV Cache compression with token selection can be formulated as attention score prediction, which is a two-dimensional (2D) time series prediction problem.** Specifically, we predict the attention in step $t+1$ as $\hat{A}_{t+1}$ and select the critical token positions with $S = TopB(\hat{A}_{t+1})$. Since stacking attention vectors over time yields a 2D spatiotemporal attention map $\mathcal{A}_t = \langle A_i \rangle_{i=1}^{t}$, where the temporal axis corresponds to decoding steps and the spatial axis corresponds to token positions in context as Figure 1. We can thus frame attention score prediction as a time series prediction problem. The input to our predictor is a 2D spatiotemporal sequence $\mathcal{A}_H = \langle A_i^{comp} \rangle_{i=t-H+1}^{t}$, representing the $H$ most recent block-wise compressed attention scores, where $A_i^{comp}$ is generated via max-pooling on the original attention score $A_i$ (detailed in Sec. 3.4). We then train a lightweight model to predict the attention scores for step $t+1$ as $\hat{A}_{t+1} = \mathcal{F}(\mathcal{A}_H)$, where $\mathcal{F}(\cdot)$ denotes the model function.

## 3.3 Attention Temporal Patterns

**Observation.** We observe that attention exhibits consistent and structured temporal behaviors, which we term **attention patterns**. At a local level, these patterns are remarkably stable. We have identified three fundamental patterns that reflect strong invariances under spatiotemporal shifts. However, these patterns reveal a complex dynamic globally, exhibiting long-term evolution. To fully understand this dynamic, we will first provide an analysis of the stable patterns.

**Three Attention Patterns.** Our work is motivated by the observation that attention exhibits three distinct temporal patterns—re-access, se-

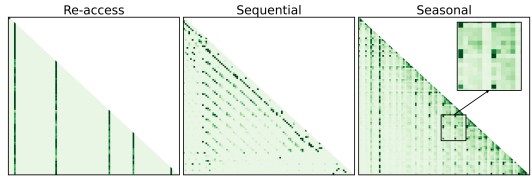

Figure 2: Visualization of three predictable temporal attention patterns. **Re-access** shows repeated attention to specific tokens. **Sequential** shows attention progresses toward the next tokens. **Seasonal** exhibits periodic recurrence as alternating bands of high and uniform attention scores.

quential, and seasonal, as shown in Figure 2. These patterns collectively suggest that attention evolves with both temporal continuity and structural regularity across tokens. We unify the three local spatiotemporal attention patterns to the following approximate invariance under joint temporal and spatial shifts:

$$a_{t,i} \approx a_{t+\delta_t, i+\delta_i}, \tag{2}$$

where and $\delta_t$ and $\delta_i$ represent temporal and spatial offsets, respectively. (1) The **re-access** pattern corresponds to $(\delta_t = 1, \delta_i = 0)$, where the model repeatedly attends to the same tokens. (2) The **sequential** pattern corresponds to $(\delta_t = 1, \delta_i = 1)$, reflecting progressive token-wise attention. (3) The **seasonal** pattern corresponds to $(\delta_t > 1, \delta_i = 0)$, indicating periodic recurrence of attention to fixed positions. While our re-access and sequential patterns resemble shapes defined in MInference [16], our time-series analysis also reveals a novel seasonal pattern, and our dynamic predictor improves prediction accuracy by 5% over fixed-template approaches (see Appendix C.10).

To better understand why attention follows spatiotemporal patterns, we investigate the underlying theoretical mechanisms. We find out that the high continuity between queries and position embedding plays a central role in shaping attention behavior.

**High Query Self-similarity.** We observe that the query embedding sequence $\{q_i\}_{i=1}^t$ exhibits strong temporal self-similarity, which is consistent with prior findings in [19]. Specifically, the one-step cosine autocorrelation is 0.87 as detailed in Appendix A.1, indicating a high degree of alignment between queries. As only the most recent KV cache is updated, **high query self-similarity indicates high stability of the attention score over consecutive time steps**, which is highly predictable.

For theoretical clarity, we define the raw attention scores as $\widetilde{A}_t = q_t k_{1:t}^\top$ omitting both the $1/\sqrt{d}$ factor and the softmax. This allows us to focus on the underlying structure of attention logits, as softmax is a monotonic function and does not alter the relative ranking of attention weights. For step $t + 1$, supposing the query vector evolves as $q_{t+1} = q_t + \Delta q$, the attention at step $t + 1$ is:

$$\widetilde{A}_{t+1} = q_{t+1} k_{1:t+1}^\top = (q_t + \Delta q)[k_{1:t} \,|\, k_{t+1}]^\top = q_t k_{1:t}^\top + \Delta q k_{1:t}^\top + (q_t + \Delta q) k_{t+1}^\top \quad (3)$$

Focusing on the first $t$ entries of $\widetilde{A}_{t+1}$, we obtain:

$$\widetilde{A}_{t+1}[1:t] = q_t k_{1:t}^\top + \Delta q k_{1:t}^\top = \widetilde{A}_t + \Delta \widetilde{A}, \quad (4)$$

where $\Delta \widetilde{A} = \Delta q k_{1:t}^\top$. Noting that $\|\Delta q\|$ is small due to the high autocorrelation, and the key matrix $k_{1:t}$ remains relatively stable, the term $\Delta \widetilde{A}$ is of small magnitude in norm:

$$\|\Delta \widetilde{A}\| \leq \|\Delta q\| \cdot \|\mathbf{K}_i\|$$

Therefore, $\widetilde{A}_t \approx \widetilde{A}_{t+1}$ holds pointwise to a good approximation when the query exhibits high self-similarity. This implies that the relative ranking and distribution of attention weights over past tokens change slowly across decoding steps. In other words, attention exhibits temporal smoothness and inter-step consistency, which form the basis of the observed re-access and sequential patterns.

**Position Embedding.** The attention patterns are closely related to position embedding, particularly Rotary Positional Embedding (RoPE), which applies position-dependent rotations to queries and keys. In RoPE, the query and key vectors are partitioned into $d/2$ groups along dimensions and apply rotational matrices with different frequencies $\theta$ to each group. The attention weights are calculated with RoPE encoding as [17, 42]:

$$\begin{aligned}
\text{RoPE}(q_i)^\top \text{RoPE}(k_j) &= \sum_{m=1}^{d/2} \langle q_i^{(m)}, R((j-i)\theta_m)\, k_j^{(m)} \rangle \\
&= \sum_{m=1}^{d/2} \|q_i^{(m)}\|\, \|k_j^{(m)}\| \cos\left(\phi^{(m)} + (j-i)\theta_m\right),
\end{aligned} \quad (5)$$

where $q_i^{(m)}, k_j^{(m)} \in \mathbb{R}^2$ denote the components of $q_i$ and $k_j$ in the $m$-th group, $\theta_m$ is the frequency parameter, and $\phi^{(m)}$ is the angle between $q_i^{(m)}$ and $k_j^{(m)}$. For the query and key are stable across steps, the formulation reveals that **the inner product depends primarily on the *relative position* $j - i$**. When $j - i = \Delta$ is fixed, the cosine terms remain stable across decoding steps, producing consistent attention along diagonals, showing as the **sequential pattern**. Furthermore, the periodicity of the cosine function explains the periodic slashes of sequential patterns observed in Figure 2.

**Dynamic Patterns.** These foundational patterns are not static and exhibit dynamic evolution even within a single generation sequence. As the decoding process progresses, patterns can shift, appear, or fade as illustrated in Figure 2. This dynamism is driven by the decay of query-key similarity from inherent query shifting, and is especially evident in long-output tasks like reasoning.

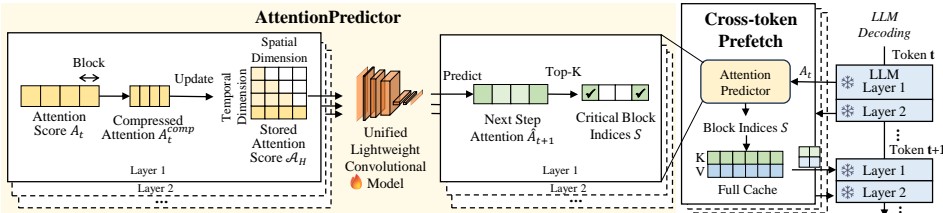

Figure 3: Overview of AttentionPredictor and cross-token prefetching framework. (a) **Attention-Predictor** formulates the history attention scores as a spatiotemporal sequence, and predicts the attention at the next step with a pre-trained model. To enhance efficiency, the attention history is updated in a compressed form at each decoding step. (b) **The cross-token prefetching framework** asynchronously evaluates critical tokens and fetches KV for the next token during the LLM inference, thereby accelerating the decoding stage.

**Predict Attention with Time Series Methods.** Since the query similarity and position embedding are inherent features of LLMs and independent of input, the resulting attention pattern is also inherent to LLMs. This allows a single shared predictor that can generalize across heads and datasets. For the pattern also changes as the sequence grows, this dynamic requires the predictor not only to recover the stable structures but also to adapt to shifts. Building on theoretical insights into attention, we propose that the pattern can be treated as a two dimensional spatiotemporal process suited to time series methods. In the next section, we introduce an architecture that uses this view to capture and forecast the evolution of attention over time.

## 3.4 AttentionPredictor: A Spatiotemporal Predictor

**Overall Process.** As shown in Figure 3 and Algorithm 1, AttentionPredictor prepares an attention history sequence $\mathcal{A}_H$ in the prefilling stage, and predicts attention during the decoding stage. First, the $A_t$ from the LLM is compressed to $A_t^{comp}$ using block-wise attention compression. Next, $\mathcal{A}_H$ is updated with $A_t^{comp}$. The next step attention $\hat{A}_{t+1}$ is then predicted with the pretrained model $\mathcal{F}$. From $\hat{A}_{t+1}$, the top-K positions are selected with a budget of $B/b$, since $\hat{A}_{t+1}$ is in compressed form. Finally, the indices are expanded with $b$ to obtain the final critical token indices $S$.

**Model Design.** To capture spatiotemporal features, we introduce a convolutional neural network (CNN) composed of two 2D convolution layers followed by a 1D convolution layer. The 2D convolutions capture spatiotemporal features at multiple scales, while the 1D convolution focuses on the time dimension, extracting temporal patterns across time steps. By replacing the fully connected layer with a 1D convolution kernel, the model adapts to the increasing spatial dimension, without data segmentation or training multiple models. Compared to an auto-regressive LSTM [43], the CNN is more lightweight and offers faster predictions, main-

---

**Algorithm 1** Identify Critical Tokens

**Input**: Attention scores $A_t$, Attention history series $\mathcal{A}_H$, Block size $b$, KV budget $B$
**Output**: Critical KV token indices set $S$

1: Pad $A_t$ to the nearest multiple of $b$ with zero
2: $A_t^{comp} \leftarrow$ MaxPooling$(A_t, b)$
3: $\mathcal{A}_H \leftarrow$ Concat$(\mathcal{A}_H[-H+1:], A_t^{comp})$
4: $\hat{A}_{t+1} \leftarrow \mathcal{F}(\mathcal{A}_H)$
5: $S_{comp} \leftarrow$ Top-K$(\hat{A}_{t+1}, B/b)$
6: $S \leftarrow \bigcup_{i \in S_{comp}} \{i \cdot b, i \cdot b+1, \ldots, i \cdot b+(b-1)\}$
    **Return** KV cache indices set $S$

---

taining a prediction time shorter than the single-token inference latency. Additionally, when compared to a MLP [44] on time-series dimension, the CNN is more effective at capturing spatial features, which improves prediction accuracy.

**Training Data and Strategy.** Our model is both data-efficient and generalizable. We train the model only on a small subset of attention data, specifically approximately 3% extracted from the dataset. The model performance on the entire dataset shows our model effectively captures the patterns (see Section 4.2). Additionally, the temporal patterns of attention are inherent in LLMs, a single model can generalize well across various datasets. For example, our model trained on LongBench also performs well on the GSM8K dataset, highlighting the generalization capability of AttentionPredictor.

**Block-wise Attention Compression.** To speed up prediction, we apply attention compression before computation. By taking advantage of the attention's locality, AttentionPredictor predict attention and identify critical tokens in blocks. Inspired by Quest [31], we use the maximum attention value in each block as its representative. Specifically, max-pooling is applied on $A_t$ with a kernel size equal to the block size $b$, as $A_t^{comp} = \text{Maxpooling}(A_t, b)$, reducing prediction computation to roughly $1/b$.

**Distribution Error Calibration.** Due to the sparsity of attention computation, the distribution of attention history $\mathcal{A}_H$ used for prediction may deviate from the distribution of dense attention. This deviation tends to accumulate over decoding, particularly as the output length increases. To mitigate this issue and enhance prediction accuracy, we introduce a distribution error calibration technique to correct these deviations. Specifically, we store the full attention score every $M$ steps, effectively balancing accuracy with computational efficiency. Note that the full attention is only used to update the history. During LLM decoding, we still use the sparse KV cache, strictly adhere to the budget. Additionally, we predict and update the position $S$ every several steps for efficiency.

### 3.5 KV Cache Cross-token Prefetching

To address the increased memory cost of longer contexts, current LLM systems offload the KV cache to the CPU, but I/O transfer latency becomes a significant bottleneck in inference. KV cache prefetching offers a solution by asynchronously loading the sparse cache in advance. We introduce the *cross-token* KV cache prefetching framework, which differs from the cross-layer method [19] by leveraging longer transfer time budgets and enhancing data integration. They are helpful for improving prediction accuracy with larger models and improving transfering efficiency. Specifically, our implementation involves a prefetching process for each layer. As illustrated in Figure 3, during the decoding phase, AttentionPredictor forecasts the critical token indices $S$ for the next step. The framework then prefetches the sparse KV cache with $S$ for the next step from the full cache to the GPU. At the same time, LLM inference of other layers is also ongoing. Subsequently, the prefetched sparse KV cache is used to calculate the sparse attention of the next token. The timeline of cross-token prefetching can be seen in Figure 10 in Appendix B.1.

## 4 Experimental Results

### 4.1 Settings

**Tasks.** 1) We use the **LongBench** [45], **InfiniteBench** [46] and **RULER QA** [47] dataset for long context evaluation. LongBench is a widely used benchmark for long-context LLM inference. It consists of 16 datasets covering 6 tasks, including single- and multi-document QA (SQA and MQA), summarization, few-shot learning, synthetic tasks, and code completion. The metrics for each task are in Appendix B.3. InfiniteBench has up to extreme 124K average context length. 2) We employ the **AIME** [48] dataset, a tough mathematics dataset, for evaluating the performance of our method with the long-reasoning output scenario. 3) We use **GSM8K** math dataset to evaluate the long n-shot CoT task. 4) We use **MMLU** [49] and **GPQA** [50] dataset to evaluate the multi-choice task. 5) We use the **Needle In A Haystack** [51] experiment to evaluate the in-context retrieval capabilities.

**Baselines and LLMs.** We select four sparse attention approaches as our baselines. These include four cache eviction methods, StreamingLLM [12] and its advanced method Cascading [52], accumulative approaches H2O [11] and SnapKV [13], and two cache retrieval method Quest [31] and learning-based SeerAttention [14]. We use the official implementations of all baseline experiments. A detailed description of each baseline is provided in Appendix B.2. We choose two widely used long-context models for our evaluation: LongChat-v1.5-7b-32k [53] with 32K context length and LLaMA-3.1-8B-Instruct [54] with 128K context length.

**Implementation details.** We set the history step H to 64, the block size b to 16, and the calibration step M to 5. Performance analysis of these hyperparameters is discussed in Section 4.6. Follow Quest [31], we did not apply our method or any other algorithms to the first two layers of the LLM. Following the settings of H2O and StreamingLLM, We allocated the budget equally to the prefix and local tokens, assigning 64 tokens each. The remaining KV budget is allocated to intermediate tokens, determined by the prediction model. We conducted experiments on NVIDIA A800 (80GB) GPUs.

Table 1: The evaluation results on the LongBench dataset across 1K, 2K, and 4K KV cache budgets.

| Budget | Method | SQA | MQA | Summary | Few-shot | Synthetic | Code | Average↑ | SQA | MQA | Summary | Few-shot | Synthetic | Code | Average↑ |
|---|---|---|---|---|---|---|---|---|---|---|---|---|---|---|---|
| | | | | Longchat-v1.5-7b-32k | | | | | | | | LLaMA-3.1-8B-Instruct | | | |
| **Full** | **Full cache** | 31.07 | 23.95 | 24.70 | 63.80 | 15.25 | 54.86 | 36.06 | 43.59 | 46.00 | 26.14 | 68.78 | 52.29 | 52.51 | 48.17 |
| **1024** | StreamingLLM | 22.89 | 19.84 | 21.94 | 60.65 | 3.75 | 53.23 | 30.84 | 31.73 | 37.51 | 21.98 | 64.69 | 51.25 | 51.26 | 42.35 |
| | Cascade | 22.66 | 21.10 | 19.91 | 59.84 | 6.00 | 9.76 | 24.80 | 35.61 | 38.85 | 23.93 | 62.98 | 51.25 | 42.58 | 42.16 |
| | H2O+ | 26.98 | 22.38 | 22.92 | 61.10 | 14.00 | 54.67 | 33.58 | 37.00 | 42.28 | 24.32 | 60.81 | 54.28 | 52.21 | 44.25 |
| | SnapKV | 28.46 | 23.85 | 22.98 | 62.75 | 14.00 | **55.84** | 34.63 | 40.62 | 43.30 | 24.78 | 67.14 | 53.81 | 52.57 | 46.37 |
| | Quest | **30.31** | 21.41 | **24.60** | 63.40 | **17.50** | 52.47 | 35.29 | 42.00 | 43.87 | **26.03** | 66.89 | **54.53** | 48.87 | 46.92 |
| | AttentionPredictor | 30.18 | **24.28** | 24.20 | **63.81** | 14.00 | 54.38 | **35.62** | **43.59** | **45.88** | 25.87 | **69.42** | 53.59 | **54.69** | **48.58** |
| **2048** | StreamingLLM | 24.79 | 20.44 | 23.62 | 61.52 | 5.75 | 52.62 | 31.99 | 34.39 | 36.61 | 24.63 | 64.80 | 45.13 | 44.99 | 41.64 |
| | Cascade | 24.97 | 22.84 | 20.79 | 62.28 | 7.00 | 10.33 | 26.46 | 38.43 | 41.33 | 24.46 | 65.30 | 52.25 | 43.08 | 43.98 |
| | H2O+ | 27.51 | 22.91 | 21.51 | 62.58 | 14.50 | 54.66 | 34.09 | 40.55 | 43.78 | 24.94 | 65.51 | 53.36 | 53.58 | 46.34 |
| | SnapKV | 29.77 | 24.27 | 23.71 | 63.53 | 14.25 | **56.69** | 35.50 | 42.77 | 44.79 | 25.31 | 67.19 | 53.32 | **54.35** | 47.44 |
| | Quest | **31.64** | 23.90 | **24.51** | 64.43 | 15.75 | 53.13 | 36.14 | **43.73** | 44.75 | 25.91 | **69.39** | 54.25 | 50.01 | 48.00 |
| | AttentionPredictor | 30.91 | 24.37 | 24.34 | **64.60** | 15.25 | 54.52 | **36.15** | 42.78 | **45.25** | **26.09** | 68.26 | **55.38** | 51.81 | **48.04** |
| **4096** | StreamingLLM | 29.63 | 20.01 | 23.64 | 56.18 | 7.75 | 55.36 | 32.36 | 41.63 | 38.82 | 24.44 | 58.95 | 41.15 | 52.27 | 42.63 |
| | Cascade | 27.94 | 22.83 | 21.40 | 63.68 | 8.25 | 10.29 | 27.57 | 41.80 | 44.33 | 25.34 | 66.08 | 51.59 | 43.58 | 45.57 |
| | H2O+ | 30.07 | 23.72 | 21.74 | 63.16 | 14.75 | 54.66 | 35.06 | 43.14 | **45.21** | 25.52 | 66.14 | 53.29 | **53.61** | 47.45 |
| | SnapKV | 30.52 | 24.35 | 24.54 | 63.80 | 14.50 | **56.70** | 35.99 | 42.87 | 45.34 | **25.79** | 67.63 | **54.09** | 53.39 | 47.84 |
| | Quest | 30.94 | 22.98 | 24.61 | **64.19** | 15.50 | 53.62 | 35.83 | 43.46 | 44.51 | 25.77 | 68.23 | 53.21 | 50.16 | 47.59 |
| | AttentionPredictor | 30.78 | **24.50** | **24.65** | **64.19** | 14.75 | 54.85 | **36.10** | **43.87** | 44.57 | 25.69 | **69.07** | 53.38 | 53.44 | **48.17** |

## 4.2 Main Results

**Results on LongBench.** We evaluate our method on various long-context tasks in the LongBench benchmark, with the KV cache budgets ranging from 1024 to 4096. We restrict H2O's attention to the past 64 steps to ensure fairness and denote this variant as H2O+. As shown in Table 1, AttentionPredictor surpasses the performance of all SOTA KV cache eviction and retrieval methods across various KV budgets and LLMs. Notably, the average performance loss compared to the full cache is less than 0.5% across all cache budgets. This demonstrates the ability of our method to effectively model attention patterns and precisely predict the locations of critical tokens. Furthermore, in an extremely sparse budget setting of **8%** (1K/13K), our approach achieves full cache performance, while all other methods suffer at least a 1.25% reduction. Additional results of SeerAttention, long-context benchmark InfiniteBench and RULER QA are in Appendix C.2, C.3, and C.4.

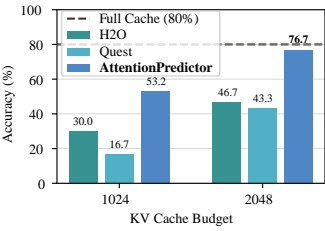

Figure 4: Evaluation results on the long-output reasoning task AIME2024 with QwQ-32B.

Figure 5: Per-token decode latency of LLaMA-3.1-8B under varying tokens, comparing ours with FlashAttention2 (full cache) and H2O.

**Results on CoT reasoning.** Reasoning significantly enhances the problem-solving capabilities of LLMs. For instance, QwQ-32B achieves an accuracy of 76.7% on the challenging AIME benchmark with the aid of detailed reasoning. However, generating complex answers leads to **output length** of up to **31.5K** tokens. This sharply contrasts with typical long-context scenarios, which involve long inputs but short outputs. Such a shift introduces distinct challenges for KV cache compression during decoding. Under this demanding setting, we evaluate AttentionPredictor against two decoding-phase acceleration baselines, H2O and Quest. As shown in Figure 4, AttentionPredictor consistently outperforms both baselines. It achieves comparable accuracy of **76.7%** with a KV budget ratio of **0.13** (2K/15K average), whereas Quest drops sharply to 43.3%. Moreover, our approach attains this accuracy with an average output of only 15K tokens, offering significantly higher efficiency compared to H2O (21K) and Quest (20K). Additional results of GSM8K, multi-choice benchmarks MMLU and GPQA are in Appendix C.1, C.5, and C.6.

### 4.3 Needle In A HayStack

We conduct the "Fact Retrieval Across Context Lengths" (Needle In A Haystack) experiment. Our approach achieved a perfect **100% retrieval accuracy** with a 64K-token context and a **1/64** cache compression ratio, surpassing all existing methods. This outcome underscores the effectiveness of our method in preserving critical information necessary for accurate retrieval.

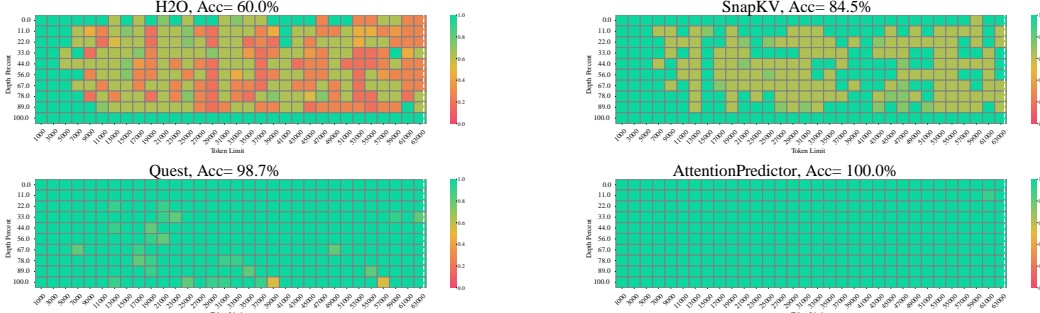

Figure 6: Results of the Fact Retrieval Across Context Lengths ("Needle In A HayStack") test in LLaMA-3.1-8B-Instruct with **64k** context size in **1024** KV cache size.

### 4.4 Efficiency

**Decode latency.** We evaluate decoding latency in a cache offloading setup [55], essential for long contexts where the KV cache exceeds GPU memory. In this scenario, the standard approach (FlashAttention2) creates a bottleneck by prefetching the entire KV cache from CPU to GPU, which our method optimizes. Further details on the setup are in Appendix B.4. As shown in Figure 5, our method consistently outperforms both FlashAttention2 and H2O across all context lengths. Notably, at 32K tokens, we achieve a **5.6×** acceleration compared to FlashAttention2, demonstrating the dramatic acceleration enabled by loading only a small fraction of the cache. Moreover, we attain a **1.6×** speedup over the calculation-efficient H2O, showing the effectiveness of our cross-token prefetch system.

**Overload breakdown.** Table 2 details the per-layer overhead of our prefetching system. A key result is that this total overhead remains remarkably stable, increasing only from 47.3ms to 50.0ms as context length grows from 4K to 32K. This stability is a core advantage of our asynchronous design. The growing latency from *Predict Attention* and *Transfer Cache* is effectively hidden by being absorbed into the *Wait for Main LLM* period, during which the main model is busy processing the current token's inference. Due to parallel execution across layers, this per-layer overhead is equivalent to the final per-token latency.

Table 2: Per-layer overhead breakdown (in ms) of our cache prefetching system, averaged across all layers.

| Context Length | 4K | 8K | 16K | 32K |
|---|---|---|---|---|
| **Predict Attention** | 0.7 | 1.2 | 2.3 | 4.5 |
| **Transfer Cache** | 2.3 | 3.9 | 7.6 | 13.2 |
| **Wait for Main LLM** | 44.4 | 43.4 | 39.7 | 32.3 |
| **Total Per-Layer Overhead** | 47.3 | 48.6 | 49.6 | 50.0 |

### 4.5 Prediction Accuracy

**Compare with Baselines.** We evaluate the prediction accuracy to evaluate our predictor, and the metric is $R_{rec}^{pred}/R_{rec}^{target}$. On the three representative tasks—QA, summary, and mathematical reasoning—with different KV cache budgets, AttentionPredictor consistently achieves a higher average recovery rate compared to H2O and Quest, as shown in Table 3. This demonstrates that our method accurately identifies the positions of critical tokens, thereby minimizing information loss. Notably, with the KV budget of 512, AttentionPredictor shows a significant advantage over Quest, achieving a 7% higher average recovery rate, emphasizing our robustness under the extremely high (20×) compression ratio.

**Generalization across datasets.** We evaluate our model's generalization by training exclusively on a single dataset and then evaluating prediction accuracy on all remaining, unseen tasks within LongBench. As shown in Figure 7, the model consistently achieves over 95% accuracy across every task, demonstrating strong cross-task transferability and generalization.

Table 3: The attention prediction accuracy (%) across different cache sizes.

| Llama-3.1-8B | | | | | |
|---|---|---|---|---|---|
| KV Budget | Method | QA | Summary | Math | Average |
| 512 | H2O | 85.90 | 87.89 | **93.89** | 89.23 |
| | Quest | 86.84 | 80.15 | 83.15 | 83.38 |
| | AttentionPredictor | **91.07** | **88.30** | 93.40 | **90.93** |
| 1024 | H2O | 89.79 | 89.92 | 95.40 | 91.70 |
| | Quest | 91.87 | 87.26 | 89.29 | 89.47 |
| | AttentionPredictor | **94.95** | **91.72** | **96.10** | **94.25** |
| 2048 | H2O | 93.82 | 92.61 | 97.09 | 94.51 |
| | Quest | 95.69 | 92.35 | 94.27 | 94.10 |
| | AttentionPredictor | **97.36** | **94.63** | **97.84** | **96.61** |
| 4096 | H2O | 97.37 | 95.65 | 98.75 | 97.26 |
| | Quest | 98.16 | 96.10 | 97.83 | 97.36 |
| | AttentionPredictor | **98.86** | **97.07** | **99.04** | **98.33** |

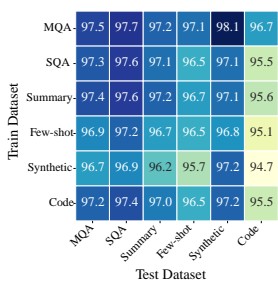

Figure 7: The prediction accuracy (%) of different dataset from LongBench.

## 4.6 Ablation Study

The effects of hyperparameter block size on AttentionPredictor performance are studied. Other hyperparameters, predictor structure and training efficiency are shown in App. C.8, C.7, and D.1.

**Block size** $b$. Figure 8 shows that AttentionPredictor maintains superior average performance with block sizes 8-64. While larger block sizes degrade performance from coarser token positioning, AttentionPredictor declines more mildly than Quest's block-wise KV cache retrieval. Quest's performance drop may stem from losing total attention score information when its min-max based per-block attention score upper bound estimation is used for retrieval, causing inaccurate block identification. In contrast, AttentionPredictor accurately captures attention patterns, improving critical block identification and mitigating block-wise retrieval drawbacks.

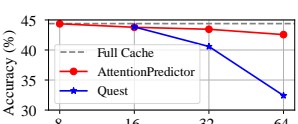

Figure 8: Evaluation of Block Size on Longchat with LongBench.

## 5 Conclusion

We present AttentionPredictor, the first learning-based method to directly predict attention patterns for KV compression and attention sparsity method. With a lightweight convolution model, AttentionPredictor captures spatiotemporal patterns of attention score and predicts the next-token attention score accurately. Additionally, we propose the first cross-token prefetching framework, which effectively mitigates prediction and transfer delays in the LLM decoding stage. Experiments on the long-context datasets demonstrate that AttentionPredictor achieves 13× KV cache compression and 5.6× speedup with comparable accuracy of LLM.

## 6 Acknowledgments

This work was supported in part by National Key R&D Program of China under contract 2022ZD0119801, National Nature Science Foundations of China grants U23A20388, 62021001, U19B2026, and U19B2044. This work was supported in part by Huawei as well. We would like to thank all the anonymous reviewers for their insightful comments. This research was also supported by the advanced computing resources provided by the Supercomputing Center of the USTC.

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

# A High Similarity of Queries and Keys

## A.1 Qeury Similarity

Our model predicts the attention at step $t + 1$ from a recent history. Since the temporal dimension corresponds to the query sequence (one query per step), the similarity between adjacent queries ($q_t$ and $q_{t+1}$) is the most critical factor for demonstrating step-by-step temporal continuity. Therefore, we evaluate the similarity of consecutive queries. Specifically, we extract query data from LLM using LongBench and compute the cosine autocorrelation of adjacent queries with a lag of 1:

$$\rho(1) = \frac{1}{t-1} \sum_{i=1}^{t-1} \frac{\langle q_i, q_{i+1} \rangle}{\|q_i\| \cdot \|q_{i+1}\|},$$

As shown in Figure 9, the heatmap illustrates the query similarity across each layer and head of the LLM. The average similarity is 86% for Longchat and 87% for LLaMA-3.1, indicating strong continuity. Furthermore, the similarity in LLaMA is more consistent across layers, while Longchat shows higher similarity in the shallower layers.

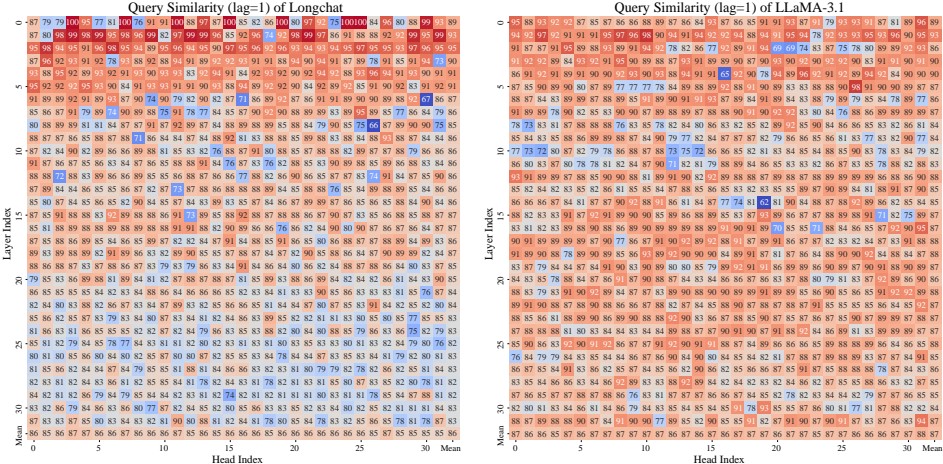

Figure 9: Query similarity of Longchat and LLaMA-3.1.

Table 4: Self similarity of query and key over different distances.

| Distance | 1 | 2 | 5 | 10 | 20 | 50 |
|---|---|---|---|---|---|---|
| Query Similarity | 0.87 | 0.83 | 0.80 | 0.78 | 0.77 | 0.76 |
| Key Similarity | 0.82 | 0.77 | 0.74 | 0.72 | 0.71 | 0.67 |

To provide a more detailed analysis of the similarity of queries at a far distance, we measure the average query cosine similarity across various distances (lags). The results are presented in Table 4. The query similarity gradually decreases as the distance increases, which is expected. However, the similarity remains relatively high even at longer distances. This finding suggests that while the influence of the most recent tokens is strongest, the attention patterns exhibit a strong local stability that decays slowly over time. This supports our model's effectiveness in capturing these evolving yet persistent patterns using a recent history window.

## A.2 Key Similarity

In addition to queries, we also investigate the similarity of keys over time. When analyzing the attention scores between pairs such as $A_{t,i}$ and $A_{t+1,i}$, both attend to the same position $i$. Consequently, the keys involved in the two computations are exactly the same ($k_i$), and therefore no issue of key self-similarity arises in this case.

In contrast, when analyzing patterns like the "Periodic Slashes" pattern, the comparison involves pairs such as $A_{t,i}$ and $A_{t+1,i+1}$, where the attention scores are influenced by different keys ($k_i$ and

$k_{i+1}$). In this case, the self-similarity of keys becomes relevant. To quantify this, we compute the cosine similarity between keys at different positions with various lags:

$$\rho(\tau) = \frac{1}{t - \tau} \sum_{i=1}^{t-\tau} \frac{\langle k_i, k_{i+\tau} \rangle}{\|k_i\| \cdot \|k_{i+\tau}\|}, \quad \tau \geq 1,$$

where $\tau$ denotes the distance between keys. The results are also presented in Table 4, and we can find that they indeed exhibit strong self-similarity. These consistently high similarity values indicate that keys vary slowly over time. Our findings are also consistent with prior works such as KIVI [34] and KVQuant [35], which reported that keys often contain fixed, massive channels, suggesting limited variation across positions.

# B    Implement Details

## B.1    Implement Details of AttentionPredictor

**Model Architecture Details.**  The architecture of the CNN-based AttentionPredictor model is summarized in Table 5. This design ensures efficiency and adaptability to varying sequence lengths, while maintaining sufficient capacity to capture temporal patterns.

Table 5: CNN-based AttentionPredictor Architecture.

| Layer No. | Layer Type | Details |
|---|---|---|
| 1 | 2D Convolution | Kernel: (3,3), Padding: 1, Channels: $1 \rightarrow 16$ |
| 2 | ReLU Activation | - |
| 3 | 2D Convolution | Kernel: (3,3), Padding: 1, Channels: $16 \rightarrow 32$ |
| 4 | ReLU Activation | - |
| 5 | Adaptive Avg. Pooling | Collapses temporal history dimension to size 1 |
| 6 | 1D Convolution | Kernel: 1, Channels: $32 \rightarrow 1$ |

**Training Details.**  We capture the attention scores during full-cache inference as raw data. We package the attention scores of $H$ steps, together with the $(H + 1)$-th step's score, as input-output pairs. The training data is block-compressed to align with the usage during LLM model inference. Specifically, the input history attention is $\mathcal{A}_H \in \mathbb{R}^{H \times \frac{t}{b}}$ padding with zero for short attentions and the output attention score is $C \in \mathbb{R}^{1 \times \frac{t}{b}}$. Attention scores with insufficient lengths are padded with zeros at the end to ensure that all data within a package has the same length. For the attention at step $t + 1$, we only predict the first $t$ positions because the last position is derived from the newly generated key, which does not need to be compressed or transferred. Notably, only the attention scores from the decoding phase are used as output data. As a result, the training dataset includes the final $H$ steps of the pre-filling attention and all attention scores from the decoding stage. To train the model, we calculate the discrepancy between the predicted and real attention scores using mean squared error (MSE) loss. The training epoch is set to 30, and we select the model with the best attention prediction accuracy as shown in Equation 1.

**Cross-token Prefetching.**  We leverage GPU parallel streams and CPU multi-threading to parallelize the token retrieval and cache transfer across different layers. Consistent with the main experiment, we maintain uncompressed caches for the first two layers to ensure higher inference accuracy. The timeline of prefetching framework is in Figure 10.

## B.2    Implement Details of Baselines

Our experiments are based on LLMs using 16-bit floating points, and we utilize FlashAttention2 [56] during the prefilling stage to reduce GPU memory consumption. Since FlashAttention2 does not output intermediate attention scores, for methods that rely on historical attention, we additionally compute partial prefill attention to obtain the scores. Our batch size is fixed at 1. For all baselines, we use the official code implementations to ensure the effectiveness of the methods. Additionally, for all methods, we skip compress the first two layers of the LLM.

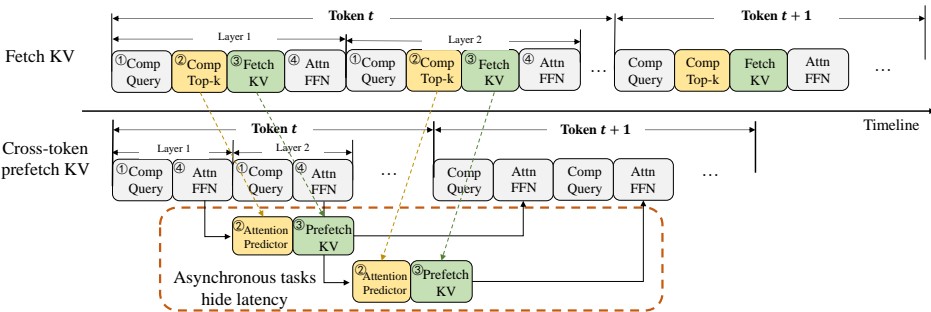

Figure 10: Timeline of our proposed cross-token prefetching. By asynchronously loading the critical KV cache for the next token, our framework hides the token evaluation and transfer latency, accelerating the decoding stage of LLM inference.

- StreamingLLM [12] finds that the sink token is very important, so it retains the initial and the most recent few tokens. For StreamingLLM, we evenly distribute the budget between sink tokens and local tokens.

- H2O [11] accumulates all historical attention scores as an estimation. We evenly distribute the budget between the heavy budget and the recent budget. Since computing the full historical data can cause OOM with longer inputs, we use the last 64 steps of historical information to calculate the heavy hitters, and denote this as H2O+. Note that accumulating all prefill attention scores causes H2O to focus on initial tokens, so this is an improvement to the method's effectiveness.

- SnapKV [13] accumulates attention scores within a recent window as an estimate and performs one-time filtering during the prefill stage. We set the time window to a size of 64 to align with our method. Since SnapKV only performs cache compression once during the prefill stage, the number of KV tokens it uses continues to grow during decoding, resulting in a relatively lower compression ratio compared to other methods.

- Quest [31] uses the current query and paged key to approximate the attention score. We use a chunk size of 16, corresponding to the block size in our method, which is also the parameter value used in the original paper.

### B.3 Details of LongBench Dataset

LongBench [45] is a carefully crafted benchmark suite designed to evaluate the capabilities of language models in processing extended documents and complex information sequences. It was developed for multi-task assessment of long-context inputs. The details of the metrics, number of words, language, and data used in LongBench are presented in Table 6.

### B.4 Details of Efficiency Experiment Setup

Our efficiency evaluation is conducted under the cache offloading scenario, which is essential for long-context generation when the KV cache size exceeds the available GPU memory. Unlike GPU-based eviction/compression, offloading keeps the full cache on CPU, preserving model accuracy. This setting is also adopted in recent works [32, 19, 33].

We conduct experiments under this offloading setup on FlashAttention2, H2O, and our proposed AttentionPredictor:

- FlashAttention2 [56]: We utilize the standard KV cache offloading implementation provided by the Hugging Face library. In this setup, at each decoding layer, the entire KV cache for the upcoming layers is prefetched from CPU to GPU to compute the full attention.

- H2O [11]: Also uses cross-layer prefetching but transfers sparse KV cache, making the comparison a more direct evaluation of the prefetching framework's efficiency.

- AttentionPredictor: Runs on an offloading setup but prefetches across tokens, improving efficiency.

Table 6: An overview of the dataset statistics in LongBench. 'Source' denotes the origin of the context. 'Avg len' refers to the average number of words, which is shorter than the token length after tokenization. 'Accuracy (CLS)' refers to classification accuracy, while 'Accuracy (EM)' refers to exact match accuracy.

| Dataset | Source | Avg len | Metric | Language | #data |
|---|---|---|---|---|---|
| *Single-Document QA* | | | | | |
| NarrativeQA | Literature, Film | 18,409 | F1 | English | 200 |
| Qasper | Science | 3,619 | F1 | English | 200 |
| MultiFieldQA-en | Multi-field | 4,559 | F1 | English | 150 |
| *Multi-Document QA* | | | | | |
| HotpotQA | Wikipedia | 9,151 | F1 | English | 200 |
| 2WikiMultihopQA | Wikipedia | 4,887 | F1 | English | 200 |
| MuSiQue | Wikipedia | 11,214 | F1 | English | 200 |
| *Summarization* | | | | | |
| GovReport | Government report | 8,734 | Rouge-L | English | 200 |
| QMSum | Meeting | 10,614 | Rouge-L | English | 200 |
| MultiNews | News | 2,113 | Rouge-L | English | 200 |
| *Few-shot Learning* | | | | | |
| TREC | Web question | 5,177 | Accuracy (CLS) | English | 200 |
| TriviaQA | Wikipedia, Web | 8,209 | F1 | English | 200 |
| SAMSum | Dialogue | 6,258 | Rouge-L | English | 200 |
| *Synthetic Task* | | | | | |
| PassageCount | Wikipedia | 11,141 | Accuracy (EM) | English | 200 |
| PassageRetrieval-en | Wikipedia | 9,289 | Accuracy (EM) | English | 200 |
| *Code Completion* | | | | | |
| LCC | Github | 1,235 | Edit Sim | Python/C#/Java | 500 |
| RepoBench-P | Github repository | 4,206 | Edit Sim | Python/Java | 500 |

# C   Additional Experimental Results on Accuracy

## C.1   Results on Long N-shot CoT GSM8K.

We evaluate the reasoning task under a long-input scenario, which is different with long output scenario in Section 4.2. As shown in Table 7, we evaluate our method on the mathematical reasoning dataset GSM8K with LLaMA-3.1.

To simulate CoT tasks within long-context, we increased the number of few-shot examples. Specifically, we randomly selected a fixed number of questions and standard CoT answer pairs as prompts, along with the questions to be tested. We chose 25, 47, and 97 few-shot examples, resulting in input lengths of approximately 4K, 8K, and 16K tokens respectively. The few-shot data were sourced from the GSM8K training set. Since the test set does not overlap with the training set, the answers remain undisclosed to the LLM.

Unlike long-context tasks LongBench, CoT mathematical reasoning tasks present distinct challenges. For example, the retrieval-based SOTA method, Quest, excels on long-context benchmarks but performs poorly on CoT tasks, showing a 16.91% accuracy drop at a sequence length of 16K. In contrast, AttentionPredictor achieves a significantly smaller accuracy loss of just 0.91%. Moreover, across all sequence lengths, our method outperforms baselines in most cases.

Table 7: Evaluation results on the long n-shot CoT GSM8K task with different input context lengths with Llama-3.1-8B-Instruct.

| | | GSM8K+LLaMA-3.1 | | | |
|---|---|---|---|---|---|
| **Method** | **Budget** | **Prompt length** | | | **Average** |
| | | **4K** | **8K** | **16K** | |
| **Full cache** | Full | 56.79 | 55.27 | 54.13 | 55.40 |
| **StreamingLLM** | 1K | 54.74 | 49.96 | 50.94 | 51.88 |
| **H2O+** | 1K | **57.16** | 52.01 | 52.16 | 53.78 |
| **Quest** | 1K | 48.52 | 45.26 | 37.22 | 43.67 |
| **SnapKV** | 1K | 53.45 | 48.67 | 49.20 | 50.44 |
| **AttentionPredictor** | 1K | 56.48 | **53.30** | **53.22** | **54.33** |

Table 8: The detailed evaluation results from LongBench of all tasks. Since the official pre-trained model of SeerAttention for LongChat is unavailable, we only evaluate its performance on LLaMA-3.1-8B-Instruct.

| Budget | Method | Single-DocumentQA | | | Multi-DocumentQA | | | Summary | | | Few-shot Learning | | | Synthetic | | Code | | Average↑ |
|---|---|---|---|---|---|---|---|---|---|---|---|---|---|---|---|---|---|---|
| | | NrtvQA | Qasper | MF-en | HotpotQA | 2WikiMQA | Musique | GovReport | QMSum | MultiNews | TREC | TriviaQA | SAMSum | PCount | PRe | Lcc | RB-P | |
| | | | | | | | | Longchat-v1.5-7b-32k | | | | | | | | | | |
| Full | Full cache | 20.70 | 29.41 | 43.09 | 33.05 | 24.14 | 14.66 | 30.84 | 22.79 | 26.61 | 66.50 | 83.99 | 40.90 | 0.00 | 30.50 | 52.94 | 56.78 | 36.06 |
| 512 | StreamingLLM | 15.03 | 17.71 | 27.14 | 27.00 | 21.52 | 9.09 | 21.51 | 19.39 | 21.97 | 55.00 | 81.07 | 37.07 | 0.50 | 5.00 | 46.40 | 53.05 | 28.65 |
| | Cascade | 16.56 | 19.02 | 24.16 | 25.57 | 22.13 | 7.88 | 11.76 | 19.50 | 15.77 | 51.50 | 82.44 | 36.73 | **1.00** | 7.50 | 6.07 | 12.45 | 22.50 |
| | H2O+ | **19.01** | 24.00 | 33.08 | 30.60 | 22.47 | 12.23 | 21.01 | 21.59 | 22.07 | 50.50 | 83.42 | 39.62 | 0.00 | 27.50 | 51.35 | 54.63 | 32.07 |
| | SnapKV | 18.73 | 25.21 | 38.18 | 33.26 | **23.79** | 13.27 | 21.84 | 21.15 | 22.58 | 59.00 | 84.20 | 38.41 | 0.00 | 25.50 | 54.39 | **57.43** | 33.56 |
| | Quest | 16.80 | **31.43** | 38.01 | 27.95 | 23.73 | 10.03 | 27.91 | **22.58** | **25.68** | 60.50 | 81.27 | 39.95 | 0.00 | **41.50** | 51.95 | 50.22 | 34.34 |
| | AttentionPredictor | 17.69 | 28.49 | **41.63** | **34.13** | 23.67 | **14.18** | **28.40** | 21.82 | 25.52 | **64.00** | **84.66** | **40.56** | **1.00** | 25.50 | **54.42** | 55.89 | **35.10** |
| 1024 | StreamingLLM | 16.30 | 20.06 | 32.30 | 28.60 | 21.37 | 9.56 | 25.52 | 20.36 | 23.51 | 60.50 | 82.15 | 39.29 | 1.50 | 6.00 | 52.74 | 53.71 | 30.84 |
| | Cascade | 19.30 | 21.21 | 27.47 | 30.24 | 22.29 | 10.78 | 14.60 | 20.63 | 19.19 | 57.00 | 83.43 | 39.10 | **2.50** | 9.50 | 6.69 | 12.83 | 24.80 |
| | H2O+ | 18.92 | 24.65 | 37.38 | 32.40 | 22.28 | 12.46 | 22.72 | 21.62 | 24.22 | 59.00 | 84.39 | 39.90 | 0.00 | 28.00 | 53.74 | 55.60 | 33.58 |
| | SnapKV | **19.46** | 26.97 | 38.96 | 34.01 | 23.22 | 14.33 | 23.17 | 21.46 | 24.50 | 64.00 | 84.33 | 39.93 | 0.00 | 28.00 | **53.96** | **57.72** | 34.63 |
| | Quest | 18.51 | 31.09 | 41.32 | 29.92 | 22.95 | 11.35 | **30.16** | **22.67** | **26.52** | 66.00 | 83.35 | 40.86 | 0.00 | **35.00** | 50.77 | 54.16 | 35.29 |
| | AttentionPredictor | 19.36 | **29.50** | **41.67** | 33.64 | **24.30** | **14.91** | 29.92 | 22.30 | 26.09 | **66.00** | **84.85** | 40.58 | 0.00 | 28.00 | 52.22 | 56.54 | **35.62** |
| 2048 | StreamingLLM | 15.52 | 22.66 | 36.20 | 29.03 | 22.15 | 10.13 | 27.59 | 21.04 | 26.20 | 64.00 | 81.93 | 38.62 | 2.00 | 9.50 | 49.78 | 55.46 | 31.99 |
| | Cascade | 19.49 | 24.80 | 30.63 | 33.48 | 22.68 | 12.37 | 16.82 | 20.82 | 20.76 | 63.00 | 83.51 | 40.32 | **3.00** | 11.00 | 7.28 | 13.38 | 26.46 |
| | H2O+ | 19.10 | 26.62 | 36.82 | 33.23 | 22.66 | 12.83 | 25.13 | 21.93 | 21.09 | 62.50 | 84.65 | 40.60 | 0.00 | 29.00 | 53.33 | 55.99 | 34.09 |
| | SnapKV | 20.02 | 28.69 | 40.61 | 34.13 | 24.02 | 14.67 | 26.05 | 21.82 | 25.59 | 65.00 | 84.71 | 40.89 | 0.00 | 28.50 | 54.93 | **58.44** | 35.50 |
| | Quest | 19.12 | **32.08** | **43.72** | 31.91 | **24.97** | **14.83** | 31.56 | **22.65** | 26.36 | 66.50 | **85.09** | 41.69 | 0.00 | **31.50** | 50.53 | 55.73 | 36.14 |
| | AttentionPredictor | **20.07** | 30.06 | 42.60 | **34.26** | 24.20 | 14.64 | 30.53 | 22.07 | **26.61** | **67.00** | 84.94 | **41.86** | 0.00 | 30.50 | 52.45 | 56.59 | **36.15** |
| 4096 | StreamingLLM | 19.45 | 27.90 | 41.53 | 28.54 | 20.50 | 10.99 | 26.86 | 20.73 | 26.54 | 58.00 | 75.71 | 34.82 | **2.50** | 13.00 | 54.31 | 56.41 | 32.36 |
| | Cascade | 20.47 | 27.98 | 35.38 | 32.45 | 23.90 | 12.15 | 17.87 | 21.33 | 21.46 | 64.50 | **84.99** | 41.55 | 0.50 | 16.00 | 7.48 | 13.09 | 27.57 |
| | H2O+ | **20.53** | 28.28 | 41.41 | 32.78 | 23.87 | 14.52 | 27.81 | 21.76 | 21.71 | 64.50 | 84.09 | 40.90 | 0.00 | 29.50 | 52.71 | 56.61 | 35.06 |
| | SnapKV | 20.30 | 28.90 | 42.36 | 33.49 | **24.46** | **15.11** | 28.37 | 22.54 | 26.54 | 66.00 | 84.19 | 41.20 | 0.00 | 29.00 | 54.70 | **58.69** | 35.99 |
| | Quest | 19.67 | 29.26 | **43.90** | 32.72 | 23.61 | 12.61 | 31.49 | **22.95** | 26.27 | **67.00** | 84.38 | 41.18 | 0.00 | **31.00** | 51.62 | 55.61 | 35.83 |
| | AttentionPredictor | 20.12 | **29.33** | 42.88 | **34.12** | 24.27 | 15.10 | 30.74 | 22.70 | **26.59** | 67.00 | 84.14 | **41.42** | 0.00 | 29.50 | 53.00 | 56.69 | **36.10** |
| | | | | | | | | LLaMA-3.1-8B-Instruct | | | | | | | | | | |
| Full | Full cache | 29.28 | 45.36 | 56.12 | 56.67 | 48.90 | 32.43 | 33.77 | 25.20 | 27.08 | 74.00 | 91.48 | 40.85 | 5.07 | 99.50 | 56.86 | 48.15 | 48.17 |
| 512 | StreamingLLM | 22.49 | 24.93 | 38.64 | 48.48 | 41.41 | **29.43** | 24.08 | 20.24 | 23.38 | 58.50 | 81.60 | 38.22 | 8.00 | 98.50 | 55.32 | 45.49 | 41.17 |
| | Cascade | 25.49 | 30.09 | 39.83 | 52.05 | 44.75 | 22.92 | 24.16 | 21.49 | 24.57 | 48.00 | 87.49 | 40.61 | 8.50 | 94.50 | 45.58 | 36.33 | 40.40 |
| | H2O+ | 25.93 | 29.59 | 45.53 | 50.59 | **45.75** | 29.24 | 24.50 | 23.28 | 23.56 | 48.50 | 88.90 | 41.39 | 8.16 | **99.50** | 56.06 | 47.88 | 43.02 |
| | SnapKV | 26.51 | 39.81 | 52.05 | 54.60 | 46.31 | 26.92 | 24.27 | 23.22 | 24.16 | 65.50 | 90.93 | 40.31 | 4.39 | 99.00 | 55.66 | 46.64 | 45.02 |
| | SeerAttention | 20.46 | 43.07 | 52.82 | 48.86 | 41.54 | 26.32 | 29.86 | 22.88 | 26.37 | 61.50 | 88.10 | **42.51** | 4.50 | 54.00 | 40.49 | 33.32 | 39.79 |
| | Quest | 25.82 | **44.01** | 53.99 | **58.08** | 43.96 | 28.84 | **32.90** | 24.50 | **26.80** | **70.00** | 88.55 | 39.42 | **11.16** | **99.50** | 51.65 | 42.03 | 46.33 |
| | AttentionPredictor | **27.53** | 43.92 | **56.58** | 57.78 | 45.30 | 27.00 | 30.21 | **24.92** | 26.54 | 67.50 | **92.28** | 40.45 | 7.50 | **99.50** | **59.11** | **46.81** | **47.06** |
| 1024 | StreamingLLM | 24.69 | 27.84 | 42.65 | 48.20 | 37.82 | 26.50 | 26.80 | 21.55 | 22.40 | 64.00 | 89.42 | 40.65 | 8.00 | 94.50 | 56.18 | 46.33 | 42.35 |
| | Cascade | 27.14 | 37.17 | 42.51 | 53.06 | 37.75 | 25.73 | 26.74 | 22.25 | 25.61 | 58.50 | 89.70 | 40.73 | 6.50 | 96.00 | 46.88 | 38.27 | 42.16 |
| | H2O+ | 26.24 | 36.30 | 48.47 | 53.97 | 45.78 | 27.08 | 26.11 | 23.59 | 25.05 | 53.00 | 88.54 | 40.88 | 9.05 | 99.50 | 55.20 | 49.22 | 44.25 |
| | SnapKV | 27.63 | 41.86 | 52.37 | 55.83 | 46.39 | 27.69 | 26.39 | 24.06 | 25.49 | 70.00 | 90.66 | 40.76 | 8.12 | 99.50 | 55.55 | 49.58 | 46.37 |
| | SeerAttention | 23.64 | 46.13 | 54.41 | 54.61 | 38.78 | 28.13 | 31.47 | 24.28 | **26.69** | 65.50 | 89.21 | 41.79 | 6.00 | 82.00 | 45.97 | 35.49 | 43.38 |
| | Quest | 27.46 | 43.58 | 54.95 | **57.22** | 44.53 | 29.86 | **33.64** | **25.37** | 26.68 | 72.00 | 88.61 | 40.05 | **9.55** | 99.50 | 53.92 | 43.82 | 46.92 |
| | AttentionPredictor | **28.30** | **46.66** | 55.81 | 56.87 | **50.67** | 30.10 | 32.34 | 25.08 | 26.66 | **74.00** | **92.11** | **42.15** | 7.17 | **100.00** | **59.12** | **50.26** | **48.58** |
| 2048 | StreamingLLM | 25.79 | 32.35 | 45.03 | 49.84 | 36.71 | 23.29 | 29.31 | 22.43 | 26.83 | 69.00 | 86.87 | 38.53 | 7.25 | 83.00 | 43.32 | 46.66 | 41.64 |
| | Cascade | 28.49 | 40.74 | 46.05 | 51.86 | 45.54 | 26.58 | 28.98 | 22.42 | 26.49 | 63.50 | 90.55 | 41.85 | **10.50** | 94.00 | 47.03 | 39.13 | 43.98 |
| | H2O+ | 28.07 | 40.35 | 53.22 | 54.81 | 44.75 | 31.77 | 28.20 | 23.50 | 26.37 | 64.50 | 90.71 | 41.32 | 6.71 | **100.00** | 57.97 | 49.19 | 46.34 |
| | SnapKV | 28.00 | 45.34 | 54.98 | **57.97** | 45.89 | 30.50 | 28.92 | 24.29 | 26.32 | 71.50 | 89.36 | 40.71 | 6.64 | **100.00** | **58.79** | **49.90** | 47.44 |
| | SeerAttention | 27.56 | **46.05** | 55.70 | 55.83 | **47.07** | 29.09 | 32.02 | 24.25 | 26.82 | 68.50 | 88.24 | 40.14 | 6.00 | 93.50 | 46.94 | 37.74 | 45.34 |
| | Quest | 29.25 | **46.05** | **55.90** | 57.91 | 45.51 | 30.82 | **34.05** | **24.89** | **26.93** | **74.00** | 92.41 | 41.77 | 8.50 | **100.00** | 54.29 | 45.72 | 48.00 |
| | AttentionPredictor | **29.52** | 45.72 | 54.20 | 56.50 | 46.93 | **33.45** | 33.28 | 24.68 | 26.88 | 72.00 | **91.79** | **42.20** | 10.08 | 99.50 | 57.47 | 49.09 | **48.33** |
| 4096 | StreamingLLM | 25.88 | 43.45 | 55.55 | 49.90 | 37.78 | 28.79 | 28.13 | 22.18 | 26.70 | 62.00 | 78.53 | 36.32 | 8.29 | 74.00 | **58.63** | 45.91 | 42.63 |
| | Cascade | 28.43 | 46.21 | 50.77 | 55.02 | 47.68 | 30.29 | 31.47 | 23.75 | **26.93** | 65.50 | 90.03 | 41.70 | 8.67 | 94.50 | 47.14 | 40.02 | 45.57 |
| | H2O+ | 28.37 | 45.33 | 55.72 | 56.27 | 47.74 | **31.63** | 30.91 | 24.15 | 26.89 | 65.50 | **90.96** | 41.97 | 7.08 | 99.50 | 57.52 | **49.70** | 47.45 |
| | SnapKV | 29.17 | 44.03 | 55.40 | 56.14 | **49.01** | 30.86 | 31.36 | 24.73 | 26.85 | 72.00 | 90.31 | 40.58 | 8.68 | 99.50 | 57.75 | 49.03 | 47.84 |
| | SeerAttention | **30.43** | 44.06 | **56.00** | 55.67 | 45.73 | 31.08 | 32.60 | 24.49 | 26.82 | 68.50 | 89.44 | 42.94 | **10.62** | 99.50 | 46.13 | 39.46 | 46.59 |
| | Quest | 29.30 | 46.43 | 54.66 | 55.88 | 47.01 | 30.65 | **34.54** | **25.23** | 26.31 | 71.00 | 89.76 | **43.92** | 6.42 | **100.00** | 55.25 | 45.06 | 47.59 |
| | AttentionPredictor | 28.67 | **47.07** | 55.87 | **57.46** | 48.35 | 27.91 | 33.12 | 24.76 | 26.62 | **73.50** | 90.30 | 43.40 | 6.75 | **100.00** | 58.03 | 48.85 | **48.17** |

## C.2 Additional Results on Longbench Benchmark

Due to space constraints, we presented the test results after aggregating task types in Section 4.2. Here, we present the individual results for all tasks. The Table 8 shows that our method surpasses the majority of the SOTAs on most tasks, and the average performance under all budgets and LLMs exceeds all compared methods, demonstrating the effectiveness of our approach.

SeerAttention utilizes a learnable module to aid cache compression. We use the officially released SeerAttention model for our tests. Since the official pre-trained model of SeerAttention for LongChat is unavailable, we evaluate its performance on LLaMA-3.1-8B-Instruct.

Table 8 indicates that our method outperforms SeerAttention, which also employs a learnable module. In particular, with a cache budget of 512, our method achieves an average accuracy 7.27% higher than SeerAttention. This could be attributed to SeerAttention encoding keys and queries without directly modeling attention scores, thus failing to accurately retrieve important token blocks under extreme resource constraints. Moreover, by employing a single unified model for all LLM layers, our

approach results in a total model size that is merely 0.02% of SeerAttention's, further attesting to the effectiveness of our method.

## C.3 Results on InfiniteBench

To evaluate the effectiveness of our proposed method, we conduct experiments on selected tasks from the InfiniBench [46] benchmark, an extremely long-context benchmark with an average context length 214K. We use LLaMA-3-8B as the base model and focus on the Math Find and Choice tasks, which require numerical reasoning and multi-option selection, respectively. We compare our approach against several state-of-the-art KV compression baselines under an 8K token KV cache budget.

As shown in Table 9, AttentionPredictor achieves the highest average performance (44.9), outperforming Quest (43.4), SnapKV (42.5), and InfLLM (33.7). Notably, on the Math Find task, which is particularly sensitive to the preservation of precise attention information, our method attains 34.3—significantly higher than all baselines. These results suggest that AttentionPredictor is more effective at identifying and preserving the most relevant attention information under constrained memory, enabling better retention of long-range dependencies.

Table 9: Evaluation on InfiniteBench.

| InfiniteBench with Llama-3-8B | | | | |
|---|---|---|---|---|
| KV Budget | Method | Math Find | Choice | Average |
| Full | Full cache | 21.7 | 44.1 | 32.9 |
| 8K | InfLLM [32] | 23.7 | 43.7 | 33.7 |
| 8K | SnapKV [13] | 27.4 | **57.6** | 42.5 |
| 8K | Quest [31] | 32.3 | 54.5 | 43.4 |
| 8K | **AttentionPredictor** | **34.3** | 55.5 | **44.9** |

## C.4 Results on RULER QA Benchmark

We further evaluate our method on the RULER QA benchmark [47], which is designed to assess long-context understanding under varying context lengths and constrained KV cache budgets. The results are summarized in Table 10. The results demonstrate that with a highly constrained budget of only 1K tokens, our AttentionPredictor consistently and significantly outperforms the SnapKV baseline across all tested context lengths.

Table 10: Evaluation results on the RULER QA benchmark with different input context lengths with Llama-3.1-8B-Instruct.

| RULER QA Benchmark with Llama-3.1-8B-Instruct | | | | |
|---|---|---|---|---|
| Method | KV Budget | Context Length | | |
| | | 4K | 8K | 16K |
| **Full cache** | Full | 75.0 | 73.0 | 69.0 |
| **Snap KV** | 1K | 63.7 | 71.7 | 68.0 |
| **AttentionPredictor** | 1K | 73.4 | 71.9 | 68.8 |

## C.5 Results on MMLU Benchmark

To further assess the effectiveness of our approach, we conduct experiments on the MMLU benchmark [49]. Since the default evaluation mode in the official lm-eval library is a single-step prediction task, it does not fully reflect the benefits of our method, which is designed to optimize multi-step decoding. Therefore, we adopt the cot_fewshot configuration, which applies an n-shot Chain-of-Thought (CoT) prompting strategy and involves a multi-step reasoning process. This setup provides a more suitable testbed for evaluating our decoding-stage optimizations.

As shown in Table 11, our proposed AttentionPredictor achieves performance very close to that of the full-cache model while consistently outperforming the SnapKV baseline under different KV cache

budgets. These results demonstrate that our method remains effective even on relatively short input tasks, provided that the decoding process requires multi-step reasoning.

Table 11: Evaluation results on the MMLU benchmark with few-shot CoT.

| MMLU Benchmark with Llama-3.1-8B-Instruct | | |
|---|---|---|
| **Method** | **Budget** | |
| | 512 | 1024 |
| **Full cache** | 67.17 | 67.17 |
| **Snap KV** | 65.60 | 66.17 |
| **AttentionPredictor** | 66.27 | 66.64 |

## C.6 Results on GPQA Benchmark

To further validate our approach on expert-level tasks, we also tested it on the GPQA benchmark [50]. GPQA is a challenging dataset of graduate-level multiple-choice questions curated by domain experts across biology, physics, chemistry, and philosophy. It is similar in format to MMLU but significantly more difficult, requiring deep expert-level reasoning.

As shown in Table 12, our method again proves effective, maintaining high accuracy in a scenario requiring deep reasoning.

Table 12: Evaluation results on the GPQA benchmark with n-shot CoT.

| Method | Budget | 5-shot | 10-shot | 15-shot | Average |
|---|---|---|---|---|---|
| **Llama-3.1-8B-Instruct** | Full | 31.31 | 37.88 | 31.31 | 33.50 |
| **SnapKV** | 1K | 22.73 | 29.80 | 23.23 | 25.25 |
| **AttentionPredictor** | 1K | 31.31 | 33.33 | 31.82 | 32.15 |

## C.7 Prediction Model

Our prediction task can be framed as modeling spatio-temporal information inherent in the token sequences, a problem for which various architectures have been explored [43, 44, 57, 58]. We evaluate the performance of various prediction model implementations. MLP is applied to the sequence dimension to handle variations in token length. For LSTM, attention is divided into 16 width blocks and predictions are independent for each block, referred to as LSTM-block in Table 13. Similarly, CNN-block utilizes a block-wise prediction approach, employing a fully connected layer for fixed block lengths. Finally, the CNN model, as the primary setting in our experiments, predicts all attention scores simultaneously. As reported in Table 13, CNN outperforms other models, achieving the highest attention prediction accuracy by effectively capturing attention patterns. MLP failed to account for neighboring token interactions, while LSTM-block and CNN-block were restricted to block-level information without global context. Notably, CNN also required the fewest parameters and was the most memory-efficient during inference.

Table 13: Prediction accuracy and parameter numbers of different model implementations.

| LongBench+Longchat | | |
|---|---|---|
| Model type | Total params | Recovery rate |
| MLP | 49K | 92.88 |
| LSTM-block | 3.2M | 91.75 |
| CNN-block | 21K | 91.94 |
| **CNN** | **4.9K** | **93.65** |

## C.8 Ablation Study Results of Hyperparameters

We evaluate the impact of hyperparameters on our method. Specifically, we train the predictor using the default parameters of the main experiment, i.e., $b = 16$, $H = 64$. We employ full attention without sparse computation during training, which is equivalent to performing distribution error calibration at each step, i.e., $M = 1$. Subsequently, we test the performance under different hyperparameters using the trained model. Other parameters during testing remain consistent with the main experiment, with a KV budget of 1K. We evaluate six representative tasks in Longbench. The results are shown in Table 14.

**Calibration step** $M$**.** Table 14 shows the average performance of AttentionPredictor with calibration steps ranging from 1 to 20. Performance improves with shorter calibration intervals, indicating that the calibration scheme reduces cumulative errors caused by differences between sparse and original attention distributions. However, higher calibration frequencies increase computational costs, creating a trade-off between accuracy and efficiency.

**History step** $H$**.** As depicted in Table 14, we evaluate the average accuracy of AttentionPredictor over a range of history steps. Overall, the performance gap compared to the full cache remains consistently small. Notably, the middle value of $H$ maximizes performance by providing the necessary information for pattern recognition, while mitigating redundancy caused by the decaying self-correlation of attention scores.

Table 14: Results of AttentionPredictor with different hyperparameters.

| | | LongBench+Longchat | | | | | | |
| Hyperprameter | | SQA | MQA | Summary | Few-shot | Synthetic | Code | Average |
| | | MF-en | HotpotQA | QMSum | TriviaQA | Pre | Lcc | |
| **Full Cache** | | 43.09 | 33.05 | 22.79 | 83.99 | 30.50 | 52.94 | **44.39** |
| **History Step** | **16** | 41.83 | 34.00 | 22.22 | 84.45 | 28.00 | 51.87 | **43.73** |
| | **32** | 41.60 | 33.90 | 22.32 | 84.81 | 30.00 | 52.53 | **44.19** |
| | **64** | 41.67 | 33.64 | 22.30 | 84.85 | 28.00 | 52.22 | **43.78** |
| | **128** | 41.51 | 34.39 | 22.45 | 84.35 | 26.00 | 52.96 | **43.61** |
| **Calibration Step** | **1** | 42.89 | 33.27 | 22.35 | 84.70 | 28.50 | 52.53 | **44.04** |
| | **2** | 42.47 | 33.61 | 22.43 | 84.90 | 28.00 | 52.13 | **43.92** |
| | **5** | 41.67 | 33.64 | 22.30 | 84.85 | 28.00 | 52.22 | **43.78** |
| | **10** | 41.10 | 33.87 | 22.37 | 84.85 | 28.00 | 52.36 | **43.76** |
| | **20** | 41.54 | 33.99 | 22.12 | 84.90 | 28.00 | 52.50 | **43.84** |
| **Block Size** | **8** | 43.09 | 34.61 | 22.38 | 85.29 | 28.00 | 52.76 | **44.36** |
| | **16** | 41.67 | 33.64 | 22.30 | 84.85 | 28.00 | 52.22 | **43.78** |
| | **32** | 40.85 | 34.47 | 22.46 | 84.87 | 25.00 | 53.02 | **43.45** |
| | **64** | 42.60 | 34.04 | 22.01 | 84.85 | 19.00 | 52.86 | **42.56** |

## C.9 Necessity of the Learnable Attention Predictor

A pertinent question arises regarding the necessity of a learnable attention predictor: given the potential for high similarity between attention distributions of adjacent tokens or layers, **could one simply utilize the attention map from a preceding token or layer as a direct estimate?** To address this, we conducted a comparative experiment evaluating our proposed AttentionPredictor against two such heuristic baselines: (1) **Previous-Layer Attention**, which uses the attention map from the corresponding position in the immediately preceding layer, and (2) **Previous-Token Attention**, which employs the attention map of the immediately preceding token within the same layer.

Table 15: Performance comparison on LongBench with Llama3.1-8B-Instruct.

| | | Single-DocumentQA | | | Multi-DocumentQA | | | Summary | | | Few-shot Learning | | | Synthetic | | Code | | Average↑ |
| Budget | Method | NrtvQA | Qasper | MF-en | HotpotQA | 2WikiMQA | Musique | GovReport | QMSum | MultiNews | TREC | TriviaQA | SAMSum | PCount | PRe | Lcc | RB-P | |
| Full | Full cache | 29.28 | 45.36 | 56.12 | 56.67 | 49.90 | 32.43 | 33.77 | 25.20 | 27.08 | 74.00 | 91.48 | 40.85 | 5.07 | 99.50 | 56.86 | 48.15 | 48.17 |
| 1024 | **Previous layer** | 26.04 | 33.93 | 46.17 | 53.50 | 46.04 | 26.80 | 26.27 | 21.94 | 25.81 | 64.00 | 83.48 | 35.74 | **10.75** | 99.50 | 48.60 | 41.58 | 42.51 |
| 1024 | **Previous token** | 28.90 | 39.84 | **55.37** | 55.63 | 48.66 | 30.30 | 27.36 | 23.45 | 25.73 | 69.50 | 88.95 | **43.01** | 8.50 | **99.56** | **56.96** | 47.40 | 46.82 |
| 1024 | **AttentionPredictor** | 28.50 | **47.35** | 54.99 | 57.53 | 45.91 | **31.63** | **32.52** | **24.51** | **26.78** | **73.00** | 89.54 | 42.12 | 6.49 | **99.56** | **56.96** | 47.43 | **47.80** |

The results of this evaluation, performed on the LongBench benchmark using Llama3.1-8B-Instruct, are presented in Table 15. Employing previous-layer attention yields an average score of 42.51%, which is a substantial 5.29% lower than our AttentionPredictor's average of 47.80%. This performance decrement underscores the limitation of direct cross-layer attention transference, primarily because attention patterns often exhibit considerable variation across different layers, making the previous layer's attention a less reliable proxy for identifying critical tokens in the current layer.

The previous-token attention strategy achieves a higher average score of 46.82%. This is 4.31% better than the previous-layer approach and aligns with the observation that attention mechanisms frequently display sequential patterns along the token dimension within the same layer. However, despite this improvement, relying solely on the single immediately preceding token is still suboptimal.

Our learnable AttentionPredictor, with an average score of 47.80%, outperforms the previous-token attention baseline by 0.98%. This superiority stems from our model's capacity to learn and leverage more complex, dynamic attention patterns from several previous tokens, rather than being confined to the heuristic of the single last token, which may not adequately capture evolving contextual dependencies.

Therefore, these comparative results affirm that **while attention similarities exist, simplistic heuristic estimations are insufficient**. The empirical evidence highlights the necessity and distinct advantage of our proposed learnable AttentionPredictor for achieving more accurate anticipation of attention distributions.

## C.10 Comparison between MInference and AttentionPredictor

To further evaluate the effectiveness of our proposed AttentionPredictor, we compare its performance against MInference [16]. MInference operates by categorizing attention heads into three predefined sparse patterns (A-shape, Vertical-Slash, and Block-Sparse) and uses a kernel-aware search to assign the optimal pattern to each head offline. During inference, it dynamically builds sparse indices based on the assigned pattern and the specific input.

We consider two variants of MInference for a comprehensive comparison: **(1) MInference (prefill)**: The standard version where the pattern-fitting algorithm is run only during the pre-filling stage. **(2) MInference (decode)**: A stronger, more dynamic variant where the computationally expensive pattern-fitting algorithm is re-run at every decoding step to adapt to new contexts.

As shown in Table 16, AttentionPredictor consistently outperforms both MInference variants across all tasks, achieving an average accuracy of 94.25%. This demonstrates that by learning from recent history instead of relying on fixed pattern templates, AttentionPredictor more accurately captures the dynamic nature of attention during decoding.

Table 16: Attention prediction accuracy (%) on Llama-3.1-8B-Instruct with 1K KV budget.

| Method | QA | Summary | Math | Average |
|---|---|---|---|---|
| MInference (prefill) | 90.51 | 84.45 | 93.47 | 89.48 |
| MInference (decode) | 92.18 | 88.74 | 94.67 | 91.86 |
| **AttentionPredictor** | **94.95** | **91.72** | **96.10** | **94.25** |

## C.11 Top-K Selection Strategy Analysis

**Why Top-K Outperforms Sampling?**  While prior works such as MagicPIG [59] suggest that Top-K may introduce bias in certain tasks, our experiments reveal a critical insight: **prediction accuracy matters more than the design of the selection policy.** As shown in Table 17, we compare the performance of our AttentionPredictor and MagicPIG on the LongBench benchmark using Llama-3.1-8B-Instruct (MagicPIG's results are taken from the original paper). The results demonstrate that when combined with our high-precision AttentionPredictor, even a simple Top-K strategy achieves superior performance compared to MagicPIG's more sophisticated sampling approach. This indicates that high-quality attention predictions can enable basic selection policies to outperform more complex alternatives. Furthermore, it is important to note that our AttentionPredictor is decoupled from the selection policy and **can be flexibly integrated with various methods**, including both Top-K and importance sampling.

Table 17: Longbench task results on Llama-3.1-8B-Instruct with 10% KV budgets.

| Method | Qasper | RB-P | Lcc | Pre | TREC | TriviaQA | Average |
|---|---|---|---|---|---|---|---|
| **Full Cache** | 45.36 | 48.15 | 56.86 | 99.50 | 74.00 | 91.48 | 69.22 |
| **MagicPig** | 43.96 | 46.45 | 57.06 | 100.00 | 74.40 | 91.38 | 68.87 |
| **AttentionPredictor** | 45.50 | 50.26 | 59.41 | 99.67 | 74.00 | 88.61 | 69.58 |

# D  Additional Experimental Results on Efficiency

## D.1  Training Efficiency

We evaluate the impact of the training data ratio on prediction accuracy on the hotpotqa dataset of Longbench. The predictor's performance improved only slightly, from 96.0% to 97.5%, as the proportion of training samples increased from 3% to 70%. This efficiency can be attributed to two factors. First, the predictor only needs to capture the relative patterns rather than precisely predict attention scores, which is easier. Second, since attention patterns are inherent characteristics of the LLM, the predictor exhibits strong generalization capabilities. These findings suggest that a relatively small amount of training data is adequate for the predictor to effectively learn the underlying attention dynamics.

Table 18: Accuracy of predictor with different training sample ratios. The test set is a 20% split from the datasets.

| Training Samples Ratio | 0.03 | 0.05 | 0.1 | 0.2 | 0.5 | 0.7 |
|---|---|---|---|---|---|---|
| Accuracy (%) on Test Samples | 96.0 | 96.2 | 95.9 | 96.6 | 97.2 | 97.5 |

## D.2  Computational Overhead of AttentionPredictor.

We have quantified the inference overhead of our predictor model. As shown in Table 19, both the latency and memory footprint are minimal, ensuring that the prediction process does not become a bottleneck.

Table 19: The computational overhead of the prediction model.

| Context Length | Prediction Latency (ms) | Peak Memory Usage (MB) |
|---|---|---|
| 4K | 0.28 | 97 |
| 8K | 0.47 | 194 |
| 16K | 0.91 | 388 |
| 32K | 2.22 | 776 |

## D.3  Memory Efficiency of Cross-Token Prefetching

We have evaluated the memory footprint of our proposed cross-token prefetching framework during decoding. As shown in Table 20, the framework consistently consumes less GPU memory compared to the standard cross-layer approach. This advantage arises because the memory savings from loading only a small, sparse KV cache outweighs the potential overhead from the cross-token prefetching mechanism.

Table 20: Peak GPU memory consumption (GB) of LLaMA-3.1-8B during decoding.

| Context Length | Standard Cross-Layer | Our Cross-Token | Memory Reduction |
|---|---|---|---|
| 4K | 32.9 | 30.1 | 8% |
| 8K | 35.9 | 30.3 | 16% |
| 16K | 41.9 | 30.5 | 27% |
| 32K | 53.9 | 31.1 | 42% |

# E  Limitations

Our current investigation is primarily centered on a specific range of LLM architectures and sizes, and the generalizability of observed attention patterns to vastly different model types warrants further study. The necessary conditions for a specific attention pattern can be explored in the future work.

## F   Boarder Impacts

Academic Impact: Our study of LLM attention score patterns and their predictability may offer insights into attention dynamics, potentially informing future research on these characteristics. By exploring attention score patterns, this work could also be relevant to advancements in KV cache compression, possibly contributing to the development of more memory-efficient model designs.

Societal Impact: Through the potential implications for KV cache compression via attention predictability, this research could contribute to more efficient LLM inference. This might lead to reduced latency in AI applications and lower energy consumption. Such improvements could potentially enhance the accessibility of LLMs and support more sustainable AI practices.

## G   License

We include the following licenses for the code, models, and benchmarks we used in this paper.

Benchmarks: LongBench [45]: License, AIME 2024 [48]: License, GSM8K [60]: License, Needle In A Haystack [51]: License, InfiniteBench [46]: License. Models: LLaMA-3.1-8B-Instruct [54]: License, LongChat [53]: License. Code: H2O [11]: License, Minference [16]:License, SeerAttention [14]: License.

