# OpenReview forum: "AttentionPredictor: Temporal Patterns Matter for KV Cache Compression"
_NeurIPS.cc/2025/Conference — NeurIPS 2025 poster_

### Official Review · Reviewer_8ZdG · 2025-06-28

**Clarity:** 3
**Significance:** 2
**Originality:** 3
**Rating:** 4
**Confidence:** 3

**Summary:**

The paper presents a design named AttentionPredictor, a learning-based method for Key-Value (KV) cache compression in large language models (LLMs), enabling efficient long-context inference. It utilizes a lightweight, unified convolutional model to capture spatiotemporal patterns in attention scores, predicting next-step attention scores, and integrates a cross-token prefetching framework to mitigate prediction and cache loading latencies.

**Questions:**

See weaknesses above. Also other minors:

Sections C.5 and 4.6 provide standard deviations but do not specify the error computation method (e.g., closed-form, bootstrap) or distributional assumptions (e.g., normality). Please provide these details.

Calibration step M and history step H (Table 9) are empirically determined (lines 631-647). Please provide a theoretical justification or sensitivity analysis for their impact on performance.

The cross-token prefetching framework claims to hide latency (Figure 5), but lacks a detailed breakdown of overhead components (e.g., prediction time, data transfer, synchronization). Can you quantify the contribution of each component to latency and explore potential optimizations?

**Ethical Concerns:**

["NO or VERY MINOR ethics concerns only"]

**Final Justification:**

Thank the authors for the submission and the feedback during the rebuttal. Also, thanks for the other reviewers' comments, which all help the reviewer to finalize the evaluation (keep positive for the study).

**Limitations:**

yes

**Quality:**

3

**Strengths And Weaknesses:**

Strengths:

The solution introduces a learning-based approach for predicting direct attention scores and a cross-token prefetching framework, which differs from heuristic-based methods and per-layer models. Its lightweight unified prediction model enhances scalability and efficiency.

It also reduces memory and computational demands for long-context tasks, with implications for multi-turn dialogues, complex reasoning, and resource-constrained settings.

The paper articulates the theoretical basis, model design, and implementation details.

Weaknesses:

While standard deviations are provided, the paper does not specify the method for error computation or the distributional assumptions. Experiments are limited to LLaMA-3.1-8B and QwQ-32B, lacking validation on other architectures.

While societal impacts, such as reduced latency and energy consumption, are noted, no deployment data for edge devices is provided.
Hyperparameter choices are empirically determined without theoretical justification or sensitivity analysis. The communication overhead analysis for cross-token prefetching lacks specific latency data.

---

> ### Author Rebuttal · Authors · 2025-07-30
>
> We thank the reviewer for the insightful and valuable comments. We respond to each comment as follows and sincerely hope that our rebuttal could properly address your concerns. If so, we would deeply appreciate it if you could raise your score. If not, please let us know your further concerns, and we will continue actively responding to your comments and improving our submission.
> ### **Weaknesses and Questions**
>
> > W1-1 & Q1: While standard deviations are provided, the paper does not specify the method for error computation or the distributional assumptions. Sections C.5 and 4.6 provide standard deviations but do not specify the error computation method (e.g., closed-form, bootstrap) or distributional assumptions (e.g., normality). Please provide these details.
>
>
> Thank you for this point. To ensure the statistical significance of our results, we conducted each experiment five times and reported the mean and standard deviation of the outcomes.
>
> The standard deviation was calculated using the common formula for **sample standard deviation**, which provides a robust measure of performance variance without assuming a specific underlying data distribution. The formula used is:
>
> $s = \sqrt{\frac{\sum_{i=1}^{n}(x_i - \bar{x})^2}{n-1}}$,
>
> where $s$ is the sample standard deviation, $n$ is the number of experimental runs, $x_i$ is the result of each individual experiment, and $\bar{x}$ is the arithmetic mean of the five experimental results.
>
> This method accurately reflects the variability in our results. We will add this clarification to the revised manuscript.
>
> > W1-2: Experiments are limited to LLaMA-3.1-8B and QwQ-32B, lacking validation on other architectures.
>
> We appreciate the suggestion. As detailed in **Section 4.1** of our paper, the evaluation was conducted on three distinct models: **LongChat-v1.5-7b-32k, LLaMA-3.1-8B-Instruct, and QwQ-32B**. The results presented in **Table 1, Table 4, and Figure 4** demonstrate that AttentionPredictor is effective across these different model families. We agree that further validation is valuable and, we plan to extend our evaluation to other popular architectures like Mistral in our future work.
>
> > W2-1: While societal impacts, such as reduced latency and energy consumption, are noted, no deployment data for edge devices is provided.
>
> Thank you for this question. Our work primarily targets **large-scale, cloud-based LLM inference systems**, where the immense memory footprint of the KV cache for long contexts is a critical bottleneck. While adapting KV cache compression for resource-scarce edge devices is an important research direction, it presents a different set of challenges and is beyond the primary scope of our current paper. We have clarified this target application scope in the manuscript.
>
>
> > W2-2 & Q2: Hyperparameter choices are empirically determined without theoretical justification or sensitivity analysis. Calibration step M and history step H (Table 9) are empirically determined. Please provide a theoretical justification or sensitivity analysis for their impact on performance.
>
> We thank the reviewer for this point. We provide a sensitivity analysis for these key hyperparameters in **Section 4.6 and Appendix C.5**, with detailed results in **Table 9**. Our paper provides the following justification for these choices:
> * **For Calibration Step (M):** As discussed in the appendix on lines 638-642, there is a clear trade-off. More frequent calibration improves accuracy by reducing cumulative errors from distribution drift, but at the cost of higher computational overhead.
> * **For History Step (H):** As explained on lines 643-647, a moderate value for H is optimal because it provides sufficient historical information for pattern recognition, while a larger value can introduce redundancy due to the decaying self-correlation of attention scores.
>
> > W2-3 & Q3: The communication overhead analysis for cross-token prefetching lacks specific latency data. The cross-token prefetching framework claims to hide latency (Figure 5), but lacks a detailed breakdown of overhead components (e.g., prediction time, data transfer, synchronization).
>
> Thank you for your insightful comment. Our cross-token prefetching framework is designed to hide this latency by overlapping asynchronous tasks with the main LLM computation. The process is illustrated in the timeline in **Figure 10 (Appendix B.1)**. The key components are:
> * **Prediction Time:** The prediction overhead is negligible because our unified CNN model is extremely lightweight at only 21KB.
> * **Data Transfer & Synchronization:** The primary latency comes from fetching the KV cache blocks. This is hidden by executing the transfer in parallel with the computations of other layers of the *current* token.
>
> Table 4 shows a detailed breakdown. Note that since we perform attention prediction once every two tokens, the average per-token inference latency is further reduced.
>
> **Table 4:** Computation time breakdown with 1/16 budget. All values are reported in milliseconds per token (**ms/token**).
>
> Context Length|Attention Prediction|Cache Transfer|Sync/Wait|Total Overhead
> -|-|-|-|-
> 4K|1.4|2.3|73.7|77.3
> 8K|2.3|3.9|72.3|78.6
> 16K|4.6|7.6|67.3|79.6
> 32K|9.1|13.2|57.7|80.0

---

> > ### Comment · Reviewer_8ZdG · 2025-08-06
> >
> > Dear authors,
> >
> > Thank you for providing the rebuttal, which addresses most of my concerns raised in the initial review. The clarification on the standard deviation calculation, the correction regarding the number of models evaluated, and the justified scoping of the work to cloud-based systems are all well-received. The provided justifications for hyperparameter choices are grounded in empirical sensitivity analysis and clear trade-offs. A follow-up question arises from the data provided in the rebuttal's Table 4. I still remain positive about the submission.
> >
> > Some minor questions for the rebuttal. Could you please elaborate on the system dynamics that cause the Sync/Wait time to decrease as context length increases? Is the observed constant total overhead a fundamental benefit of the cross-token prefetching framework?

---

> ### Author Response · Authors · 2025-08-08
> **Further Response to Reviewer 8ZdG**
>
> Dear Reviewer 8ZdG,
>
> Thank you very much for your positive and constructive feedback. We sincerely appreciate your recognition of our efforts in addressing the concerns raised in the initial review, especially regarding the clarification of the standard deviation calculation, correction of model counts, and the scoping to cloud-based systems. We’re also grateful for your acknowledgment of our empirical analysis and justification of hyperparameter choices.
>
> We are happy to further clarify the follow-up questions you raised:
> ## **Response to Question: Regarding Table 4**
>
> > Could you please elaborate on the system dynamics that cause the Sync/Wait time to decrease as context length increases? Is the observed constant total overhead a fundamental benefit of the cross-token prefetching framework?
>
>
> Thank you for your question. The decrease in **Sync/Wait** time with longer context lengths is indeed a fundamental result of our **cross-token prefetching design**.
>
> As the context length increases, both **Attention Prediction** and **Cache Transfer** time grow due to larger attention maps and more KV blocks being prefetched. However, since these steps are fully **asynchronous and run in parallel with the main LLM’s computation** for the current token (t), the increased latency is absorbed into the LLM’s own inference time. This inference time remains nearly constant because the LLM operates on a compressed cache of fixed budget, **making its computational cost relatively stable across context lengths**.
>
> The **Sync/Wait** column captures the remaining idle time after prefetching finishes but before the LLM starts decoding the next token (t+1). As prediction and transfer take longer at larger context lengths, there’s **less idle time left**, hence the decreasing Sync/Wait value.
>
> This behavior illustrates a core strength of our system: **longer prefetching pipelines are accommodated without increasing the overall per-token latency**, thanks to tight pipelining with the main inference process.
>
> ***Copy of Original Table 4:** Computation breakdown at 1/16 budget (in ms/token).*
> Context Length|Attention Prediction|Cache Transfer|Sync/Wait for next token|Total Overhead
> -|-|-|-|-
> 4K|1.4|2.3|73.7|77.3
> 8K|2.3|3.9|72.3|78.6
> 16K|4.6|7.6|67.3|79.6
> 32K|9.1|13.2|57.7|80.0
>
> ---
> We hope these additional clarifications address your remaining questions. Thank you again for your thoughtful review and continued support of our work.
>
> Best regards,
>
> Authors

---

> ### Author Response · Authors · 2025-08-09
> **Eagerly await your valuable feedback**
>
> Dear Reviewer 8ZdG,
>
> We would like to express our sincere gratitude once again for your positive feedback, insightful comments, and constructive suggestions. Your guidance has been invaluable in helping us improve the quality of our work! With sincere dedication, we have carefully addressed your remaining concerns on **the overhead of our prefetch system**, as detailed in our preceding responses—in response to your invaluable comments.
>
> We are writing to gently remind you that **the author-reviewer discussion period will end in less than 12 hours**. We eagerly await your feedback **to understand if our responses have adequately addressed your concerns. If so, we would deeply appreciate it if you could raise your score**. If not, we are eager to address any additional queries you might have, which will enable us to further enhance our work.
>
> Once again, thank you for your kind support and constructive suggestions!
>
> Best,
>
> Authors

---

### Official Review · Reviewer_Y2CJ · 2025-06-30

**Clarity:** 2
**Significance:** 2
**Originality:** 2
**Rating:** 3
**Confidence:** 4

**Summary:**

This work finds Temporal Patterns in the attention scores of RoPE transformers. Based on this Temporal Patterns, this work use CNNs to predict critical tokens of time t+1 based on Attention scores patterns before t. This work computes the full attention score every M steps to avoid error accumulation. The author uses LongBench and simple Needle-in-a-HayStack tasks to evaluate the proposed method.

**Questions:**

- What is the reason of seasonal pattern?
- What is the performance of the proposed method on other tasks? e.g. RULER qa tasks, GSM8K, MMLU?

**Ethical Concerns:**

["NO or VERY MINOR ethics concerns only"]

**Final Justification:**

maintain my original score.

**Limitations:**

yes

**Paper Formatting Concerns:**

no.

**Quality:**

2

**Strengths And Weaknesses:**

## Strengths:
- The discovered Temporal Patterns in attention scores of RoPE transformers is interesting.
- This work also provides a reaonable explaination of the sequential patterns from RoPE viewpoint.
- Related works are well discussed.

## Weakness:
- The writing remains a large improvement space. E.g. line 146 mentioned the notion $A^{comp}$, but doesn't tell what is $A^{comp}$.
- Some explaination are still questionable. E.g. line 171 High query self-similarity.  We know that softmax attention is a normalized score. This author only show the high similarity (0.87) of consecutive queries. To support the explaination in line 171, one need to also show the similarity of queries at far distance. Moreover, ther author doesn't explain the reason of Seasonal Pattern.
- the calibration step M is 5, a small number. AKA, every 5 steps the proposed method needs to 1) compute attention scores on the whole sequence; 2) fetch KV cache of the whole sequence. The former requires high computation cost, while the later requires high memory cost.
- The evaluation are conducted on narrow tasks, a simple needle-in-a-haystack task and a longbench task.

---

> ### Author Rebuttal · Authors · 2025-07-30
>
> We thank the reviewer for the insightful and valuable comments. We respond to each comment as follows and sincerely hope that our rebuttal could properly address your concerns. If so, we would deeply appreciate it if you could raise your score. If not, please let us know your further concerns, and we will continue actively responding to your comments and improving our submission.
> ### **Weaknesses and Questions**
>
> > W1: The writing remains a large improvement space. E.g. line 146 mentioned the notion $A^{comp}$, but doesn't tell what is $A^{comp}$.
>
>
> Thank you for your valuable feedback on our manuscript and for highlighting areas for improvement in our writing.
> We will revise the text at its first mention.
>
> **Revision for lines 146-147:**
> "The input to our predictor is a 2D spatiotemporal sequence
> $A_H = \langle A_i^{\text{comp}} \rangle_{i=t-H+1}^t$ ,
> representing the H most recent block-wise compressed attention scores, where each $A^{comp}_i$ is generated via max-pooling on the original attention score $A_i$ (as detailed in Section 3.4)."
>
> > W2-1: Some explaination are still questionable. E.g. line 171 High query self-similarity. We know that softmax attention is a normalized score. This author only show the high similarity (0.87) of consecutive queries. To support the explaination in line 171, one need to also show the similarity of queries at far distance.
>
> Thank you for this insightful question.
>
> **1. Rationale for Consecutive Query Similarity**.
>
> Our model predicts the attention at step $t+1$ from a recent history. Since the **temporal dimension corresponds to the query sequence** (one query per step), the similarity between adjacent queries ($q_t$ and $q_{t+1}$) is the most critical factor for demonstrating step-by-step temporal continuity. This is why we initially focused on the high **consecutive query similarity (lag-1) of 0.87**.
>
> **2. On Attention Normalization**.
>
> It is also worth noting that the Softmax function normalizes attention scores **across the key dimension** for a single query; it does not influence the relationship between different queries across the temporal dimension.
>
> **3. Analysis of Query Similarity Across Distances**
>
> To provide a more detailed analysis of long-range stability, we measure the average query cosine similarity across various distances (lags). The results are presented in **Table 1**.
> The query similarity **gradually decreases as the distance increases**, which is expected. However, the similarity remains **relatively high even at longer distances**. This finding suggests that while the influence of the most recent tokens is strongest, the attention patterns exhibit a strong **local stability** that decays slowly over time. This supports our model's effectiveness in capturing these evolving yet persistent patterns using a recent history window.
>
> ***Table 1**: Query Similarity over Different Distances.*
> **Distance**|1|2|5|10|20|50
> :-|:-:|:-:|:-:|:-:|:-:|:-:
> **Similarity**|0.87|0.83|0.77|0.74|0.71|0.67
> > W2-2 & Q1: The author doesn't explain the reason of Seasonal Pattern.
>
> Thank you. The seasonal pattern arises from combining two factors:
> 1.  **RoPE's Periodicity**: Rotational embeddings are inherently periodic.
> 2.  **Contextual Periodicity**: In structured data, queries separated by a period $S$ can be similar, so $q_t \approx q_{t+S}$.
>
> The attention score is
> $A_{t,i} = \sum_m \lVert q_t^{(m)} \rVert \cdot \lVert k_i^{(m)} \rVert \cdot \cos\left( \phi_{t,i}^{(m)} + (i - t)\theta_m \right)$.
> Each frequency $m$ in RoPE has a period $T_m = 2\pi/\theta_m$. We assume the input's seasonal period $S$ aligns with a dominant frequency's period ($S\theta_{m^*} \approx 2\pi$).
>
> Consequently, the rotational component for the relative distance $(i-(t+S))$ aligns with that for $(i-t)$. When this RoPE-induced alignment combines with periodic queries ($q_t \approx q_{t+S}$) and the same key vector $k_i$, we find that **$A_{t,i} \approx A_{t+S, i}$**, which explains the seasonal pattern.
>
> > W3: the calibration step M is 5, a small number. AKA, every 5 steps the proposed method needs to 1) compute attention scores on the whole sequence; 2) fetch KV cache of the whole sequence. The former requires high computation cost, while the later requires high memory cost.
>
> We thank the reviewer for their insightful comment regarding the computational and memory overhead of the calibration step.
>
> To analyze this trade-off, we conducted **an ablation study on the calibration step M**. The results on LLaMA-3.1-8B are presented in Table 2, comparing different calibration frequencies (M=5, M=20) and no calibration (w/o M).
>
> Based on these results, we highlight three points:
>
> 1. **The calibration step is an optional enhancement.** Our core method without calibration already outperforms SOTA. This demonstrates the strength of our prediction mechanism, with calibration serving only to further refine accuracy.
> 2. **Our method offers a flexible accuracy-efficiency trade-off.** Increasing the calibration interval from M=5 to M=20 reduces the calibration computational overhead by 75% with only a minor performance decrease, still surpassing baselines.
> 3. **The calibration overhead is minimal and can be hidden.**
>    * **Low-Cost Approximate Calculation**: The calibration only requires block-level attention scores, so we can therefore use **efficient approximation methods**, similar to those in Quest, to compute with minimal computational cost.
>    * **Parallel Offloading:** In modern inference systems that separate prefill and decoding stages, this lightweight **attention calculation can be offloaded to the CPU**. This allows it to run in parallel, ensuring no impact on the main model's inference throughput.
>
> ***Table 2**: Sensitivity Analysis of Calibration Step M on Llama-3.1-8B*
> Method|Single-Doc QA|Multi-Doc QA|Summary|Few-shot|Synthetic|Code|Average
> -|-|-|-|-|-|-|-
> Full Cache|56.12|56.67|25.20|91.48|99.50|56.86|64.31
> SnapKV|52.05|54.60|23.22|90.93|99.00|55.66|62.58
> Quest|53.99|58.08|24.50|88.55|99.50|51.65|62.71
> **AttentionPredictor (w/o M)**|54.71|54.92|24.22|91.47|99.50|59.74|**64.09**
> **AttentionPredictor (M=20)**|54.71|54.92|24.62|91.47|99.50|59.75|**64.16**
> **AttentionPredictor (M=5)**|54.94|54.85|25.32|91.48|99.5|59.62|**64.29**
>
> > W4 & Q2: The evaluation are conducted on narrow tasks, a simple needle-in-a-haystack task and a longbench task. What is the performance of the proposed method on other tasks? e.g. RULER qa tasks, GSM8K, MMLU?
>
> We thank the reviewer for their insightful feedback and constructive suggestions.
> In our initial submission, beyond the Needle-in-a-Haystack and LongBench tasks, we also included evaluations on:
> * **The GSM8K benchmark with a long n-shot CoT setting**, to assess performance on long-input mathematical reasoning. This is presented in Appendix C.1.
> * **A long-output reasoning task (AIME)**, which presents distinct challenges compared to long-input tasks. The results are detailed in Section 4.2 and Figure 4.
> * **The InfiniBench benchmark**, which involves extremely long contexts with an average length of 214K tokens. These results can be found in Appendix C.3.
>
>
> To further address your comments with a more appropriate benchmark, we are conducting additional experiments on the **RULER qa benchmark**. We are providing the GSM8K results from our paper here for your convenience.
>
> **GSM8K Results (n-shot CoT)**
> As shown in table 3 below (Table 4 from our paper), AttentionPredictor maintains high accuracy on the benchmark across various long-context prompt lengths, significantly outperforming other methods.
>
> ***Table 3**: Evaluation results on the long n-shot CoT GSM8K task with Llama-3.1-8B-Instruct.*
> Method|Budget|4K|8K|16K|Average
> -|-|-|-|-|-
> **Full cache**|Full|56.79|55.27|54.13|55.40
> StreamingLLM|1K|54.74|49.96|50.94|51.88
> H2O+|1K|52.16|52.01|57.16|53.78
> Quest|1K|48.52|45.26|37.22|43.67
> SnapKV|1K|53.45|48.67|49.20|50.44
> **AttentionPredictor**|1K|**56.48**|**53.30**|**53.22**|**54.33**
>
> ### **RULER qa Results**
> Following the reviewer's suggestion, we have conducted new experiments on the RULER QA benchmark. The results are presented in **Table 4**.
>
> ***Table 4**: Evaluation results on the RULER qa benchmark under various context lengths and KV cache budgets.*
> Method|Budget|4K|8K|16K|
> -|-|-|-|-
> Llama-3.1-8B-Instruct|full|75.0|73.0|69.0
> SnapKV|1K|63.7|71.7|68.0
> **AttentionPredictor**|1K|**73.4**|**71.9**|**68.8**
>
> The results demonstrate that with a highly constrained budget of only 1K tokens, our **AttentionPredictor consistently and significantly outperforms** the SnapKV baseline across all tested context lengths.
>
> **MMLU**: We appreciate the suggestion to evaluate on MMLU. However, we believe it is **not the most suitable benchmark** to demonstrate the effectiveness of our method for two main reasons:
> 1.  MMLU tasks are resolved in a single generation step, which **only involves the prefill stage**. Since our AttentionPredictor method operates during the decoding stage, it is **not applied in these scenarios**.
> 2.  Even with a standard 5-shot setting, the average context length for MMLU tasks is only around 500 tokens. This does not align with the **long-context** scenarios our work targets, where KV cache compression is a critical challenge.
>
> We hope these clarifications and additional results adequately address your concerns. Thank you again for your valuable feedback, which has helped us strengthen our paper.

---

> > ### Comment · Reviewer_Y2CJ · 2025-08-04
> >
> > Thanksfor your reply. For the reason of seasonal pattern, you provides two reasons:
> > - RoPE's Periodicity: Rotational embeddings are inherently periodic.
> > - Contextual Periodicity: In structured data, queries separated by a period $S$ can be similar, so $q_t \approx q_{t+S}$.
> >
> > I have two questions:
> > 1) what kind of contextual periodictiy is it in practice? Could you provide an example? Is it an assumption?
> > 2) It seems contextural periodicity ($q_t \approx q_{t+S}$) is enough to produce a seasonal pattern. Is it correct?
> >
> > You mentioned that "you believe it (MMLU) is not the most suitable benchmark because MMLU is too short". Does it mean the proposed method is only appliable on the long context length scenario? Let's imagine a model incorporating  the proposed method. What do you suggest to do while the input is short?

---

> > > ### Author Response · Authors · 2025-08-07
> > > **Further Response to Reviewer Y2CJ (2)**
> > >
> > > ## **Response to Question 2: Applicability to Short Contexts**
> > >
> > > > **Reviewer's Question:** You mentioned that "you believe it (MMLU) is not the most suitable benchmark because MMLU is too short". Does it mean the proposed method is only appliable on the long context length scenario? Let's imagine a model incorporating the proposed method. What do you suggest to do while the input is short?
> > >
> > > Thank you for your question.
> > >
> > > ### **1. Evaluation on MMLU**
> > >
> > > We want to clarify that our method's applicability is **not determined by input length, but by whether the task involves a multi-step decoding phase**. Our initial testing used the `default` MMLU task in the official `lm-eval` library, which is a single-step generation task and thus not suitable.
> > >
> > > To create a testbed more suitable for our method's decoding-stage optimizations, **we now use the `cot_fewshot` configuration from the official `lm-eval` library**. This mode employs an n-shot Chain-of-Thought (CoT) prompting strategy, which involves a multi-step decoding phase, making it a valid scenario for our evaluation. The results in Table 1 show that even with a significant KV cache compression, **our AttentionPredictor closely matches the performance of the full-cache model and outperforms the SnapKV baseline**.
> > >
> > > ***Table 1**: Evaluation results on the MMLU benchmark with few-shot CoT.*
> > > | Method/Budget  | 1024 | 512 |
> > > | :--- | :---: | :---: |
> > > | Llama-3.1-8B-Instruct |67.17 |67.17 |
> > > | SnapKV |66.17 |65.60 |
> > > | **AttentionPredictor**|**66.64** |**66.27** |
> > >
> > > ### **2. Evaluation on GPQA**
> > > To further validate our approach on expert-level tasks, we also tested it on the GPQA benchmark. GPQA is a challenging dataset of graduate-level multiple-choice questions curated by domain experts across biology, physics, chemistry, and philosophy. It is similar in format to MMLU but significantly **more difficult, requiring deep expert-level reasoning**.
> > >
> > > As shown in Table 2, **our method again proves effective, maintaining high accuracy** in a scenario requiring deep reasoning.
> > >
> > > ***Table 2**: Evaluation results on the GPQA benchmark with n-shot CoT.*
> > > | Method | Budget | 5-shot | 10-shot | 15-shot | Average |
> > > | :--- | :---: | :---: | :---: | :---: |:---:|
> > > | Llama-3.1-8B-Instruct | full | 31.31 | 37.88 | 31.31 |33.50|
> > > | SnapKV | 1K | 22.73 | 29.80 | 23.23 |25.25|
> > > | **AttentionPredictor**| 1K | **31.31** | **33.33** | **31.82** |**32.15**|
> > >
> > > These combined results confirm that AttentionPredictor is effective not only in long-context tasks but also in complex, multi-step reasoning scenarios where KV cache management becomes beneficial due to CoT prompting.
> > >
> > > **We humbly hope that our responses have adequately addressed your concerns.** Furthermore, we are eager to address any additional queries you might have, which will enable us to enhance our work further.
> > >
> > > We are fully aware of your busy schedule, especially during this discussion period. We understand that reviewing numerous rebuttals can be extremely challenging. **As the deadline for the author-reviewer discussion period is approaching (due on August 8th, just one day away), and we are looking forward to your feedback and/or questions.**
> > >
> > > Thank you for your invaluable time and consideration.
> > >
> > > Best regards,
> > >
> > > Authors

---

> > > > ### Comment · Reviewer_Y2CJ · 2025-08-08
> > > >
> > > > Thanks for your response. After reading the comments, I have decided to maintain my original score.

---

> ### Author Response · Authors · 2025-08-07
> **Further Response to Reviewer Y2CJ (1)**
>
> Dear Reviewer Y2CJ,
>
> We genuinely value the opportunity to clarify and address any remaining concerns.
> ## **Response to Question 1: Seasonal Pattern Explanation**
>
> > **Reviewer's Question:** What kind of contextual periodictiy is it in practice? Could you provide an example? Is it an assumption? It seems contextural periodicity ($q_t \approx q_{t+S}$) is enough to produce a seasonal pattern. Is it correct?
>
> Thank you for the follow-up questions, which allow us to clarify the mechanism behind the seasonal pattern.
>
> ### **1. Example of Contextual Periodicity in Practice**
>
> Contextual periodicity is not a strict assumption but an **empirical observation** in structured or semi-structured inputs.
>
> A concrete example is processing a structured document like a **code**, a poem or a table.
>
> * **Example:** In a long class definition, the structure for declaring properties is often highly repetitive. The queries generated when processing each of these lines can be very similar, creating a periodic context.
> ```csharp
>       ...
>       private bool m_ReadyWait;
>       private int m_ReadyCount;
>       private bool m_Rematch;
>       public bool Rematch{ get{ return m_Rematch; } }
>       public bool ReadyWait{ get{ return m_ReadyWait; } }
>       public int ReadyCount{ get{ return m_ReadyCount; } }
>       public bool Registered{ get{ return m_Registered; } }
>       public bool Finished{ get{ return m_Finished; } }
>       public bool Started{ get{ return m_Started; } }
>       public Mobile Initiator{ get{ return m_Initiator; } }
>       ...
> ```
>
>
> ### **2. Is Contextual Periodicity ($q_t \approx q_{t+S}$) Sufficient?**
>
> You raise an excellent point regarding the role of contextual periodicity. Our analysis indicates that **contextual periodicity ($q_t \approx q_{t+S}$) is indeed the fundamental driver** behind the seasonal pattern.
>
> However, this periodicity alone is often insufficient to create the sharp, stable patterns observed in practice, because the positional shift from $t$ to $t+S$ alters the RoPE rotation for each frequency channel. The distinct seasonal pattern emerges when the input's period $S$ **resonates** with the inherent period of a dominant RoPE frequency channel ($S\theta_m \approx 2k\pi$). In this scenario, **the alignment with RoPE's periodicity makes the underlying pattern driven by the context sharp and clearly visible**.

---

> ### Author Response · Authors · 2025-08-08
>
> Dear Reviewer Y2CJ,
>
> Thank you for taking the time to review our rebuttal and for your follow-up note. While we understand and respect your decision to maintain the original score, **we would be truly grateful if you could let us know if there are any remaining concerns or points that you feel were insufficiently addressed**.
>
> We are more than happy to provide further clarification or additional analysis if needed. Your feedback is very important to us, and we genuinely hope to improve the work in any way possible based on your suggestions.
>
> Thank you again for your time and consideration.
>
> Best regards,
>
> Authors

---

### Official Review · Reviewer_gSQC · 2025-06-30

**Clarity:** 2
**Significance:** 2
**Originality:** 2
**Rating:** 4
**Confidence:** 4

**Summary:**

The paper addresses the high memory and latency costs of maintaining the full Key–Value (KV) cache during long-context decoding in large language models. It proposes AttentionPredictor, the first learning-based method for predicting future attention scores via a lightweight, shared convolutional network. The method captures patterns in past attention history to identify critical tokens for KV cache retention. When combined with cross-token prefetching, which overlaps prediction and I/O, the approach maintains accuracy close to that of the dense model under high compression, resulting in significant generation speed-ups.

**Questions:**

Clarifications:
* Line 35: Could you clarify what is meant by static patterns? My interpretation is that these are patterns constant across inputs, but methods like H2O are dynamic, selecting different critical tokens per input. If so, I don't understand this point.
* Lines 36–38: What exactly are dynamic patterns? What does it mean to say "they only represent keys or hidden states"? Based on this, I don’t fully understand why this motivates modelling the distribution of attention scores.

Requests for Additional Discussion:
* Could you more clearly describe the efficiency experiment setup, and include a comparison where offloading is not used?
* Could you explicitly discuss how token prefetching affects memory requirements?

**Ethical Concerns:**

["NO or VERY MINOR ethics concerns only"]

**Final Justification:**

Authors clarified all my concerns in the rebuttal. Thanks for that. I'm raising my score from 2 to 4. I'm keeping it at 4 because the work targets the KV off-loading setup which is more niche, therefore less significant.

**Limitations:**

No, the authors didn't properly addressed the limitations of their work which I discuss in Strengths and Weaknesses.

**Quality:**

2

**Strengths And Weaknesses:**

Strengths:
* The paper proposes an efficient, learning-based method to predict attention scores, which performs well on downstream tasks and in CoT settings.
* The authors discover a periodic seasonal pattern in attention score distributions.
* Cross-token prefetching is used effectively to hide the overhead introduced by the AttentionPredictor.

Weaknesses:

Motivation and Contributions:
* In Section 3.1, you state that "attention recovery rate reflects the amount of information preserved after compression." Could you please define information formally here? The goal of KV compression is arguably to minimize the error in model output caused by compression, which may or may not correlate with attention recovery rate. MagicPig (Chen et al., 2024), in Appendix F, discuss why Top-K is a biased estimator of attention output.
* You claim to discover temporal patterns in attention scores, but MInference, which you cite, has already identified the sequential and re-access patterns. In Lines 91–93, you write: "They struggle to capture the dynamic temporal patterns within attention scores accurately." Could you provide evidence to support this claim?
* Some theoretical explanations are unclear. In Lines 190–196, could you please further elaborate further how the formulation of RoPE leads to consistent attention along diagonals and how the periodicity of the cosine function explains the periodic slashes of the sequential patterns?

Efficiency Claims and Measurements:
* The abstract claims a 5.6× speed-up and states that the method "significantly outperforms state-of-the-arts," suggesting a general breakthrough. However, this speed-up is achieved in a specific cache-offloading setup, which is not clearly defined or explained in Section 4.3.
* The memory impact of cross-token prefetching is not discussed. My understanding is that prefetching the KV cache for the next token prediction increases GPU VRAM usage. If so, this would be unfair compared to baselines that use cross-layer prefetching.
* There is no compute breakdown provided to understand the overhead caused by the AttentionPredictor.
* The provided code is incomplete and only includes the implementation of the proposed method.

Performance Claims:
* In Line 289, the authors claim that their method surpasses the performance of other methods. However, the paper lacks statistical analysis, which is required to establish the significance of the results.
* The same issue applies to the CoT reasoning results. This is especially important given that (unless I'm mistaken) the benchmark uses a small number of samples.

Conference Checklist:
* Q4–Q5: The authors do not release the code required to reproduce baselines, which limits understanding and reproducibility.
* Q7: Standard deviations and statistical significance is not discussed.
* Q8: Experimental setup details in Section 4.1 are minimal. The only information provided is: "We conducted experiments on NVIDIA A800 (80GB) GPUs."

---

> ### Author Rebuttal · Authors · 2025-07-29
>
> We thank the reviewer for the insightful and valuable comments. We respond to each comment as follows and sincerely hope that our rebuttal could properly address your concerns.
> ## Weaknesses
> ### Motivation and Contributions
> > W1: Define information formally.
>
> We thank the reviewer for this insightful comment.
> 1. **On "Attention Recovery Rate" as a Metric:**
> We agree minimizing output error is the ultimate goal. We argue that **attention recovery rate serves as an effective proxy**, as attention output $o = A_t · V$ depends directly on the attention scores. A high recovery rate means the compressed distribution $A_t'$ closely matches $A_t$, leading to an accurate $o' \approx o$. This justifies its use to assess information preservation.
> 2. **The primary contribution is the $AttentionPredictor$**, a universal module that predicts the next-token attention distribution. It is distinct from the selection policy and **can be paired with various methods, including Top-K or importance sampling**.
> 3. **Top-K vs. Sampling & Why Our Method Excels:** While Top-K can be biased in some tasks (as noted by MagicPIG), our results show that **a precise predictor matters more than an unbiased selection policy**.
> Our AttentionPredictor captures spatiotemporal patterns to better estimate attention, so even with simple Top-K, **it outperforms MagicPIG’s complex sampling (Table 1)**, highlighting the predictor’s key role.
>
> ***Table 1**: Longbench task results with 10% KV budgets.*
> Method|Qasper|RB-P|Lcc|Pre|TREC|TriviaQA|Avg.
> -|-|-|-|-|-|-|-
> Llama-3.1-8B-Instruct|45.36|48.15|56.86|99.50|74.00|91.48|69.22
> MagicPig|43.96|46.45|57.06|100.00|74.40|91.38|68.87
> **AttentionPredictor**|45.50|50.26|59.41|99.67|74.00|88.61|**69.58**
>
>
> > W2: Provide evidence for the claim that MInference struggles to capture dynamic temporal patterns.
>
> Thank you for the thoughtful question.
> 1. **Temporal patterns are locally stable but evolve globally** during long inferences—e.g., shifting diagonal positions. Furthermore, we identified a **novel seasonal pattern**, where attention periodically shifts between token groups, a dynamic behavior not defined by MInference's categories.
> 2. **Why MInference struggles to capture** these dynamics accurately:
>    * **MInference** uses **fixed templates per head**, making it hard to adapt to such evolving or periodic structures.
>    * In contrast, our **AttentionPredictor is fully dynamic** and learns these transitions from attention history.
> 3. **Table 2** shows a **4.77% higher accuracy**, supporting our claim that modeling these dynamics leads to better attention prediction and information retention.
>
> ***Table 2**: Attention prediction accuracy (%) on Llama-3.1-8B-Instruct with 1K KV budget.*
> Method|QA|Summary|Math|**Average**
> -|-|-|-|-
> MInference|90.51|84.45|93.47|89.48
> **AttentionPredictor**|**94.95**|**91.72**|**96.10**|**94.25**
> >W3: How does RoPE lead to diagonal consistency and periodic slashes in attention patterns?
>
> Thank you for the great question.
> 1. **Consistent Attention along Diagonals**
>
> Consider the attention scores for two diagonal tokens, $A_{t,i}$ and $A_{t+1, i+1}$. Using RoPE, the score is:
> $$A_{t+1, i+1} = \sum_{m=1}^{d/2}||q_{t+1}^{(m)}||||k_{i+1}^{(m)}||\cos(\phi'^{(m)}+(i-t)\theta_{m})$$
> Note that $(i+1)-(t+1) = i - t$, so **the relative position stays the same** along the diagonal. Since RoPE encodes positions via relative angles, the cosine term is unchanged. The other terms in the sum ($||q||$, $||k||$, and the initial phase $\phi$) vary slowly across time due to self-similarity (Sec 3.3), so overall:
> $$A_{t,i} \approx A_{t+1, i+1}$$
> This leads to **stable attention scores along diagonals**.
>
> 2. **Periodic Slashes**
>
> RoPE attention scores are sums of cosine functions over relative positions:
> $$A_{t,i} = \sum_{m=1}^{d/2} C_m \cos\big(\phi^{(m)} + (i-t)\theta_m\big)$$
> Each term is periodic with period $T_m = 2\pi / \theta_m$. If the sum is dominated by a single frequency $\theta_{m*}$, **the attention score approximates a single cosine function**:
> $$A_{t,i} \approx C_{m*} \cos(\phi^{(m*)}+(i-t)\theta_{m*})$$
> This causes attention to **peak at regular intervals in key positions**, forming **periodic slash patterns** with spacing ≈ $T_{m^*}$.
>
> ### Efficiency Claims and Measurements:
> >W4 & Q3: The speed-up is not under general settings. Describe the efficiency experiment setup.
> 1. **Why Offloading Matters**: Our cache compression work **focuses on the cache-offloading scenario**, which is critical for long-context generation when KV cache exceeds GPU memory. Unlike GPU-based eviction/compression, **offloading keeps the full cache on CPU**, preserving model accuracy. This setting is also adopted in recent works [1–4].
>
> 2. **Experiment Setup.**
>     * **FlashAttention2**: Standard offloading implementation with cross-layer prefetching. At each decoding layer, it load the entire KV cache for the upcoming layers from CPU to GPU to compute the full attention.
>     * **H2O**: Also uses cross-layer prefetching but transfers sparse KV cache, making the comparison a more direct evaluation of the prefetching framework's efficiency.
>     * **AttentionPredictor**: Runs on an offloading setup but prefetches across tokens, improving efficiency.
> 3. **Applicability to General Scenarios.** While designed for offloading, AttentionPredictor is adaptable—it can act as a cache eviction policy in memory-constrained setups, predicting less important blocks to evict and speeding up decoding.
>
> [1] InfLLM: Training-Free Long-Context Extrapolation for LLMs with an Efficient Context Memory. NeurIPS 24.
>
> [2] ShadowKV: KV Cache in Shadows for High-Throughput Long-Context LLM Inference. ICML 25 (Spotlight).
>
> [3] InfiniGen: Efficient generative inference of large language models with dynamic KV cache management. OSDI 24.
>
> [4] PQCache: Product quantization-based kvcache for long context llm inference. PACMMOD 25.
>
> >W5 & Q4: The memory impact of cross-token prefetching is not discussed.
>
> Our **cross-token prefetching framework actually consumes less GPU memory** during decoding compared to a standard cross-layer approach. See **Table 3**.
> The reason is that the memory savings from loading only a **small, sparse KV cache** outweighs the potential overhead from the cross-token prefetching mechanism.
>
> ***Table 3**: Peak GPU Memory Consumption (GB) during Decoding.*
> Context Length|Standard Cross-Layer|Our Cross-Token|Memory Reduction
> -|-|-|-
> 4K|32.9|30.1|8%
> 8K|35.9|30.3|16%
> 16K|41.9|30.5|27%
> 32K|53.9|31.1|42%
> > **W6**: Lack Compute breakdown.
>
> **Table 4** shows a detailed breakdown of our cross-token prefetching process (the timeline in Figure 10, Appendix B.1). Note that since we perform attention prediction once every two tokens, the average per-token inference latency is further reduced.
>
> ***Table 4:** Computation breakdown at 1/16 budget (in ms/token).*
> Context Length|Attention Prediction|Cache Transfer|Sync/Wait for next token|Total Overhead
> -|-|-|-|-
> 4K|1.4|2.3|73.7|77.3
> 8K|2.3|3.9|72.3|78.6
> 16K|4.6|7.6|67.3|79.6
> 32K|9.1|13.2|57.7|80.0
>
> >W7 & W10: Lack baseline codes.
>
> We used the official implementations released by the original authors to reproduce the baseline results.
>
> >W8 & W9 & W11: Lack statistical analysis.
>
> We ran our method 5 times and the results are reported in **Table 5** and **Table 6**. Our method demonstrates both strong performance and consistency, outperforming other baselines.
>
> ***Table 5**: LongBench results with LLaMA-3.1-8B-Instruct (Budget=1K)*
> Method|Qasper|HotpotQA|QMSum|TriviaQA|Pre|Lcc|Avg.
> -|-|-|-|-|-|-|-
> Full Cache|45.36|56.67|25.20|91.48|99.50|56.86|62.51
> SnapKV|41.86|55.83|24.06|90.66|99.50|55.55|61.24
> Quest|43.58|57.22|25.37|88.61|99.50|53.92|61.37
> **AttentionPredictor**|**45.50 ± 0.94**|**55.40 ± 1.29**|**25.24 ± 0.13**|**91.71 ± 0.34**|**99.67 ± 0.29**|**59.41 ± 0.21**|**62.82 ± 0.35**
>
> ***Table 6**: AIME-2024 CoT reasoning task*
> Budget|Method|Acc. (%)
> -|-|-
> Full|Full Cache|80.0
> 2048|AttentionPredictor|**76.7 ± 2.8**
> >W12: Q8: Lack experimental setup details.
>
> We will provide comprehensive information on the hardware specifications (CPU: Intel(R) Xeon(R) Gold 6330 CPU @ 2.00GHz, RAM: 3.9Ti, OS: Ubuntu 22.04) and software environment (PyTorch: 2.4.0, CUDA versions: 11.8).
> ## Questions
> >Q1 & Q2: What is meant by static patterns and dynamic patterns?
>
> Thank you for your insightful questions.
> 1. **Static Patterns & H2O**:
>    * *Static patterns* are attention behaviors based on fixed rules that don’t adapt **during a single, long-output inference**.
>    * H2O uses cumulative attention scores to select tokens, favoring early, frequent ones. This helps **capture fixed-position patterns** (e.g., attention sinks) but limits responsiveness to new context later in generation.[5]
> 2. **Dynamic Patterns & SeerAttention**:
>    * *Dynamic patterns* evolve throughout generation.For instance, in a sequential pattern, the position, interval, and intensity of the diagonal lines can shift over time (as in Figure 2 in our paper).
>    * SeerAttention and similar methods use learnable modules to capture dynamic patterns. However, they **predict attention distribution indirectly**. SeerAttention, for example, learns from a compressed representation of Q&K to infer the attention distribution, not the distribution itself.
> 3. **Motivation**:
>    * This indirect approach can misalign with true attention scores. Our AttentionPredictor directly models next-token attention as a 2D time series, achieving much higher accuracy than SeerAttention—especially under tight budgets (Table 5, Appendix C.2).
>
> [5] Dialogue Without Limits: Constant-Sized KV Caches for Extended Responses in LLMs. ICML 25.

---

> ### Author Response · Authors · 2025-08-06
> **Friendly Reminder Regarding Reviewer-Author Discussion**
>
> Dear Reviewer gSQC,
>
> **We would like to once again express our sincere thanks for your insightful comments and suggestions.** Your feedback has been immensely valuable in helping us improve our work. This is a gentle reminder that there are **fewer than three days remaining** before the end of the reviewer-author discussion phase. We would greatly appreciate hearing from you if you have any further evaluations, questions, or concerns regarding our responses.
>
> *If our rebuttal has properly addressed your concerns, we would be grateful if you would consider updating your score. If not, we would be more than happy to continue the discussion and further clarify any outstanding issues.*
>
> We hope this message does not inconvenience you in any way. Thank you again for your time and thoughtful engagement.
>
> Best regards,
>
> Authors of Submission 25755

---

> > ### Comment · Reviewer_gSQC · 2025-08-06
> >
> > Excuse my late reply, and thank you for your response.
> >
> > > W2 MInference uses fixed templates per head.
> >
> > What do you mean by "fixed templates"? Are you referring to the fact that they identify three patterns and use an offline fitting procedure? If so, I agree that this is a limitation. However, their Vertical-Slash and Block-Sparse patterns are dynamic and adapt to the input.
> > Also, MInference targets prefilling, while your work focuses on decoding. What evaluation setting did you use for Table 2?
> >
> > > W3 The other terms in the sum vary slowly across time due to self-similarity.
> >
> > In Section 3.3, you only compute similarity for Q, while Line 180 mentions that the K matrix remains "relatively stable." It would be helpful to extend the empirical analysis to include self-similarity of keys as well.
> > Thanks for the detailed explanation of Periodic Slashes.
> >
> > > W4 & W5
> >
> > You're right that your method uses less memory than full cross-layer attention. At the same time, would it be correct to say that H2O with cross-token prefetching would use a similar amount of memory, and potentially be as fast—or faster—given that H2O performs less computation?
> >
> > I agree that while your method is designed for offloading, it could also be used as a cache eviction policy. In that case, what would be the computational overhead of the Attention Prediction step? Specifically, is it feasible to implement it efficiently enough to be comparable with Quest or H2O in a no-offloading setup latency-wise, while benefitting from your enhanced eviction policy?
> >
> > Could you also please explain the columns in Table 4 for computation breakdown.
> >
> > > W8
> >
> > Thanks for providing additional results.
> >
> > > Q1
> >
> > Apologies, but I didn’t fully understand your explanation. While H2O has its own biases, its method relies on attention scores, which are inherently dynamic. One could argue that your method also uses fixed rules—albeit defined by the learned Attention Predictor weights.
> >
> > > Extra Related Work
> >
> > There are two learnable KV compression methods [1, 2] that you might consider including in the related work. Given that [1] was released last year, do you still consider your method to be the first learning-based KV cache compression approach?
> > - [1] Nawrot et al., Dynamic Memory Compression: Retrofitting LLMs for Accelerated Inference
> > - [2] Łańcucki et al., Inference-Time Hyper-Scaling with KV Cache Compression
> >
> > I'm looking forward to hearing your answers, I plan to increase my score.

---

> > > ### Author Response · Authors · 2025-08-08
> > > **Further Response to Reviewer gSQC (1)**
> > >
> > > Dear Reviewer gSQC,
> > >
> > > Thank you very much for taking the time to read our rebuttal and for your thoughtful follow-up comments. **We sincerely appreciate your engagement with our work and your willingness to consider increasing your score—this means a great deal to us.**
> > >
> > > We are grateful for your insightful questions and suggestions, which help us improve the clarity and rigor of our paper. Below, we provide point-by-point responses and further clarifications to the issues you raised:
> > >
> > > ## **Response to Question 1: MInference Fix Templates**
> > >
> > > > W2 MInference uses fixed templates per head.
> > > > What do you mean by "fixed templates"? Are you referring to the fact that they identify three patterns and use an offline fitting procedure? If so, I agree that this is a limitation. However, their Vertical-Slash and Block-Sparse patterns are dynamic and adapt to the input.
> > > >
> > > > Also, MInference targets prefilling, while your work focuses on decoding. What evaluation setting did you use for Table 2?
> > >
> > > Thank you for your follow-up questions.
> > >
> > > ### 1. On "Fixed Templates" and Dynamic Patterns (W2)
> > >
> > > You've raised an excellent point. By "fixed," we refer to two aspects of MInference's design:
> > >
> > > - **Static Pattern Assignment:** Each attention head is assigned a fixed pattern type via an offline procedure.
> > > - **Predetermined Decoding Pattern:** More critically, while patterns adapt to input during the **prefill stage**, their parameters are determined **only once**.
> > >
> > > In contrast, our **AttentionPredictor is fully dynamic at every step**. It learns to predict attention from recent history without relying on predefined patterns, allowing it to capture more complex temporal dynamics.
> > >
> > > ### 2. On the Evaluation Setting
> > >
> > > You are correct that MInference targets prefilling. Our work compares methods for cache compression **during the decoding phase**. To create a fair comparison in compression ratio in the decoding stage, we adapted MInference's methodology for this context:
> > >
> > > - We first ran its official prefill analysis to identify the pattern and location for each head. We then **projected these patterns into the decoding phase** to determine key tokens (e.g., for "slash" patterns, we incremented their position by one at each step). We then compared its accuracy to our predictor using the metric from our paper's Section 4.4.
> > >
> > > ### 3. Additional Experiment: A Stronger MInference Baseline
> > >
> > > To further demonstrate the limitations of pattern-based approaches, we created a stronger baseline, **"MInference (decode)," which re-runs their computationally expensive pattern-fitting algorithm at every single decoding step**.
> > >
> > > The results in **Table 1** show **even this highly dynamic version of MInference underperforms our learning-based approach**. This reinforces our central claim: by learning from recent history instead of relying on fixed pattern templates, AttentionPredictor more accurately captures the dynamic nature of attention during decoding.
> > >
> > > ***Table 1**: Attention prediction accuracy (%) on Llama-3.1-8B-Instruct with 1K KV budget.*
> > > | Method | QA | Summary | Math | **Average** |
> > > | :--- | :---: | :---: | :---: | :---: |
> > > | MInference (prefill) | 90.51 | 84.45 | 93.47 | 89.48 |
> > > | MInference (decode) | 92.18 | 88.74 | 94.67 | 91.86 |
> > > | **AttentionPredictor** | **94.95** | **91.72** | **96.10** | **94.25** |
> > >
> > > We hope this explanation and the new results resolve the ambiguity. Thank you again for your insightful feedback.

---

> > > ### Author Response · Authors · 2025-08-08
> > > **Further Response to Reviewer gSQC (3-1)**
> > >
> > > ## **Response to Question 3: Efficiency**
> > > > W4 & W5
> > >
> > > > You're right that your method uses less memory than full cross-layer attention. At the same time, would it be correct to say that H2O with cross-token prefetching would use a similar amount of memory, and potentially be as fast—or faster—given that H2O performs less computation?
> > >
> > > > I agree that while your method is designed for offloading, it could also be used as a cache eviction policy. In that case, what would be the computational overhead of the Attention Prediction step? Specifically, is it feasible to implement it efficiently enough to be comparable with Quest or H2O in a no-offloading setup latency-wise, while benefitting from your enhanced eviction policy?
> > >
> > > > Could you also please explain the columns in Table 4 for computation breakdown.
> > >
> > > Thank you for your insightful questions. We appreciate the opportunity to clarify these points.
> > >
> > > ### **1. Potential Efficiency of H2O with Cross-Token Prefetching**
> > >
> > > Thank you for the insightful point. Indeed, if H2O were integrated into a cross-token prefetching framework similar to ours, it could potentially be faster due to its lower computational load for token selection.
> > >
> > > However, we believe **the potential speed-up for H2O might not be** as **significant** as it appears. **Our framework is designed to minimize the impact of the prediction overhead on the main inference process**. As detailed in our paper (Appendix B.1, Figure 10), the prediction and prefetching tasks are executed asynchronously and parallelized with the main LLM inference pipeline. This design effectively hides most of the analysis latency. Consequently, since the analysis latency is already largely concealed by the main inference thread, **further reducing this overhead (as would be the case with H2O) is expected to yield only marginal gains in the final end-to-end latency**.
> > >
> > > Furthermore, our system still leaves headroom for engineering improvements. The current prefetching is implemented in Python with basic threading; moving to fully asynchronous C++/CUDA kernels could yield further gains.
> > >
> > >
> > > ### **2. On the Overhead of AttentionPredictor in Non-Offloading Setups**
> > >
> > > We appreciate your interest in applying our method to non-offloading scenarios.
> > >
> > > When applied as a cache eviction policy in a non-offloading setup, **the primary acceleration comes from performing attention computation on a shorter, compressed sequence**. In this respect, different methods with the same compression budget would achieve a similar degree of speed-up from the reduced attention computation.
> > >
> > > The main differentiator becomes the computational overhead of the selection/prediction step itself. Fortunately, **our AttentionPredictor is remarkably efficient.** As demonstrated in Table 3, **the additional latency from our predictor is almost negligible compared to the per-token inference time** (below **1%**).
> > > While H2O incurs less overhead in this step, its total latency remains largely dominated by the per-token attention computation—just like ours. As a result, **the overall time cost for H2O is not significantly lower than ours**.
> > >
> > > **Therefore, even under non-offloading settings, our predictor is efficient enough to be integrated as an eviction policy without harming latency.**
> > >
> > > ***Table 3**:  A breakdown of per-token latency (ms/token) in a non-offloading (cache eviction) scenario.*
> > >
> > > - ***Full Cache Latency** measures the per-token inference time on the uncompressed context.*
> > >
> > > - ***Compressed Cache Latency** is the per-token inference time on the compressed context with budget length, showing the speed-up from reduced attention computation alone.*
> > >
> > > - ***AttentionPredictor Overhead** is the additional latency incurred by our model to predict.*
> > >
> > > - ***Total Latency (Compressed + Predictor)** represents the final, effective per-token latency of our method, calculated by summing the compressed cache latency and our predictor's overhead.*
> > >
> > > | Context Length | 8K | 16K | 32K | 64K |
> > > |-|-|-|-|-|
> > > | **Full Cache Latency (ms)** | **35.28** | **38.72** | **44.99** | **51.82** |
> > > | Budget | 512 | 1024 | 2048 | 4096 |
> > > | Compressed Cache Latency (ms) | 33.78 | 33.82 | 34.05 | 34.89 |
> > > | AttentionPredictor Overhead (ms) | 0.263 | 0.267 | 0.265 | 0.277 |
> > > | **Total Latency (Compressed + Predictor) (ms)** | **34.04** | **34.09** | **34.32** | **35.17** |

---

> > > ### Author Response · Authors · 2025-08-08
> > > **Further Response to Reviewer gSQC (3-2)**
> > >
> > > ### **3. On Computation Breakdown in Table 4**
> > >
> > >
> > > Table 4 presents the **per-token overhead** of our cross-token prefetching framework, which operates **asynchronously and in parallel with the main LLM inference** (see **Figure 10, Appendix B.1**).
> > >
> > > The columns are as follows:
> > >
> > > 1. **Attention Prediction**: Time used by our predictor to estimate next-token attention scores, after token *t* is computed.
> > > 2. **Cache Transfer**: Time to fetch the predicted critical KV blocks from CPU to GPU.
> > > 3. **Sync/Wait**: This is the "idle" time for the prefetching pipeline. During this period, **the main LLM process is busy completing the inference for the current token (t), which includes running the remaining transformer layers and the final feed-forward network** (FFN). The prefetching pipeline is essentially waiting for the LLM to generate token t and start the decoding process for token t+1, at which point the prefetched cache will be used.
> > > 4. **Total Overhead**: Sum of the above steps.
> > >
> > > While prediction and transfer time increase slightly with longer sequences, **the total overhead remains stable** (77.3ms → 80.0ms).  This is because the main bottleneck is the LLM's own inference time for the current token. **Our asynchronous, pipelined design effectively "hides" the growing prediction and transfer latency within this inference time.**As the LLM operates on a compressed budget cache, its per-token latency stays constant, ensuring the overall prefetching overhead does too.
> > >
> > > ***Copy of Original Table 4:** Computation breakdown at 1/16 budget (in ms/token).*
> > > Context Length|Attention Prediction|Cache Transfer|Sync/Wait for next token|Total Overhead
> > > -|-|-|-|-
> > > 4K|1.4|2.3|73.7|77.3
> > > 8K|2.3|3.9|72.3|78.6
> > > 16K|4.6|7.6|67.3|79.6
> > > 32K|9.1|13.2|57.7|80.0

---

> > > ### Author Response · Authors · 2025-08-08
> > > **Further Response to Reviewer gSQC (4)**
> > >
> > > ## **Response to Question 4: Dynamic H2O**
> > > > Q1
> > > > Apologies, but I didn’t fully understand your explanation. While H2O has its own biases, its method relies on attention scores, which are inherently dynamic. One could argue that your method also uses fixed rules—albeit defined by the learned Attention Predictor weights.
> > >
> > > Thank you very much for your thoughtful follow-up. We apologize for the confusion caused by our earlier explanation, and we appreciate the opportunity to clarify.
> > >
> > > You're right that **H2O reacts to each new attention score** during decoding, and in that sense, it does exhibit a form of dynamic behavior.
> > >
> > > However, our distinction between static and dynamic patterns was intended to highlight two different dimensions of adaptability:
> > >
> > > 1. **Method-level Adaptivity**:
> > >    Both H2O and our method adapt compression decisions based on intermediate attention outputs during decoding. This is indeed dynamic, and we agree that calling H2O static in this regard was misleading.
> > >
> > > 2. **Pattern-level Dynamics (Our Focus)**:
> > >    Our notion of *dynamic patterns* refers to attention behaviors that **change in structure over time**—such as evolving diagonals or shifting periodicities along the sequence dimension.
> > >
> > >    H2O, while responsive, uses a *cumulative sum* strategy that tends to emphasize early and frequent tokens. This biases the method toward fixed or repetitive patterns (e.g., attention sinks), and it lacks the capacity to model time-varying or sequential structures explicitly.
> > >
> > > In contrast, our method not only adapts decisions step-by-step but also **explicitly learns time-evolving attention patterns** by modeling the attention distribution as a 2D time series. This enables our predictor to capture richer spatiotemporal patterns beyond what can be inferred from cumulative statistics.
> > >
> > > We hope this clarifies the distinction we aimed to draw between structural adaptability (pattern-level dynamics) and reactive behavior (method-level adaptivity). Thank you again for pointing this out.
> > >
> > > ## **Response to Question 5: Extra Related Work**
> > >
> > > > There are two learnable KV compression methods [1, 2] that you might consider including in the related work. Given that [1] was released last year, do you still consider your method to be the first learning-based KV cache compression approach?
> > > >
> > > > [1] Nawrot et al., Dynamic Memory Compression: Retrofitting LLMs for Accelerated Inference
> > > >
> > > > [2] Łańcucki et al., Inference-Time Hyper-Scaling with KV Cache Compression
> > >
> > >
> > > Thank you very much for pointing this out and for suggesting additional related work.
> > >
> > > We apologize if our initial phrasing was ambiguous. To clarify, we do **not** claim to be the first *learning-based* KV cache compression method in general. Instead, as stated in our abstract and Section 1, we propose **the first learning-based KV cache compression and critical token identification method with *direct attention pattern prediction***.
> > >
> > > As you correctly pointed out, several learning-based approaches for KV cache compression already exist. These methods primarily include:
> > >
> > > * **DMC** [3][4] learn to decide whether to merge cache blocks, but they do not explicitly model or predict attention distributions.
> > > * **SeerAttention** and **NSA** [5][6] use trained models to encode keys and retrieve blocks more accurately, but these are indirect approximations based on Q/K representations.
> > >
> > > In contrast, our method:
> > >
> > > 1. **Directly models the attention score distribution** as a 2D spatiotemporal time series, enabling accurate prediction of critical tokens.
> > > 2. Requires **no fine-tuning of the LLM**, unlike DMC and NSA.
> > > 3. Uses a **unified, lightweight model** shared across all layers and heads. Our predictor is just **21KB** in size (0.02% of SeerAttention’s 101MB), making it highly scalable and memory-efficient.
> > >
> > > To prevent any future misunderstanding, we will carefully revise our manuscript. We hope this clarification addresses your concerns, and we are grateful for your willingness to reconsider your score.
> > >
> > > [3] *Dynamic Memory Compression: Retrofitting LLMs for Accelerated Inference.* ICML 24.
> > >
> > > [4] *Inference-Time Hyper-Scaling with KV Cache Compression.* arXiv preprint 2025.
> > >
> > > [5] *Seerattention: Learning intrinsic sparse attention in your llms.* arXiv preprint 2024.
> > >
> > > [6] *Native sparse attention: Hardware-aligned and natively trainable sparse attention.* ACL 25, best paper.
> > >
> > > ---
> > > We hope these additional explanations address your concerns. Once again, thank you for considering an updated score—we truly appreciate your openness and constructive critique. Your feedback is invaluable to us, and we are committed to making our work as clear, accurate, and impactful as possible.
> > >
> > > Best regards,
> > >
> > > Authors

---

> > > ### Author Response · Authors · 2025-08-09
> > > **Eagerly await your valuable feedback**
> > >
> > > Dear Reviewer gSQC,
> > >
> > > We would like to express our sincere gratitude once again for your positive feedback, insightful comments, and constructive suggestions. Your guidance has been invaluable in helping us improve the quality of our work! With sincere dedication, we have carefully addressed your remaining concerns on **MInference comparison**, **key self-similarity**, **overhead in no-offloading setup**,  **dynamic vs. fixed rules**, and **related work**, as detailed in our preceding responses—in response to your invaluable comments.
> > >
> > > We are writing to gently remind you that **the author-reviewer discussion period will end in less than 2 hours**. We eagerly await your feedback **to understand if our responses have adequately addressed your concerns. If so, we would deeply appreciate it if you could raise your score**. If not, we are eager to address any additional queries you might have, which will enable us to further enhance our work.
> > >
> > > Once again, thank you for your kind support and constructive suggestions!
> > >
> > > Best,
> > >
> > > Authors

---

> ### Author Response · Authors · 2025-08-08
> **Further Response to Reviewer gSQC (2)**
>
> ## **Response to Question 2: Self-similarity of Keys**
> > W3 The other terms in the sum vary slowly across time due to self-similarity.
> >
> > In Section 3.3, you only compute similarity for Q, while Line 180 mentions that the K matrix remains "relatively stable." It would be helpful to extend the empirical analysis to include self-similarity of keys as well. Thanks for the detailed explanation of Periodic Slashes.
> >
> Thank you for your thoughtful question. We would like to clarify two points:
>
> 1. **Regarding Line 180 and the statement about stable Keys**:
>    In this case, we are considering the attention scores between pairs such as $A_{t,i}$ and $A_{t+1,i}$, where both attend to the same position $i$. Therefore, the Keys involved in both computations are **exactly the same** ($k_i$), and thus no similarity issue arises in this context.
>
> 2. **Regarding the diagonal "Periodic Slashes" pattern and the stable Keys**:
>    This involves comparisons like $A_{t,i}$ and $A_{t+1,i+1}$, where the attention scores are influenced by different Keys ($k_i$ and $k_{i+1}$). In response to your suggestion, we **computed the similarity between Key similarity across various distances (lags) and found that they indeed exhibit strong self-similarity.** The results are presented in **Table 2**.
>
>    These high similarity values suggest that Keys vary slowly over time. Our findings are also consistent with prior works such as **KIVI** [1] and **KVQuant** [2], which observed that Keys often exhibit fixed, massive channels, indicating limited variation across positions.
>
>    ***Table 2**: Key Self-Similarity at Different Distances*
>
>    | **Distance**   | 1    | 2    | 5    | 10   | 20   |
>    | -------------- | ---- | ---- | ---- | ---- | ---- |
>    | **Similarity** | 0.82 | 0.77 | 0.74 | 0.72 | 0.71 |
>
>    [1] *KIVI: A Tuning-Free Asymmetric 2-Bit Quantization for KV Cache.* ICML 2024.
>
>    [2] *KVQuant: Towards 10 Million Context Length LLM Inference with KV Cache Quantization.* NeurIPS 2024.
>
> We hope this additional empirical evidence strengthens the explanation of the self-similarity hypothesis.

---

> ### Author Response · Authors · 2025-08-09
> **Eagerly await your valuable feedback**
>
> Dear Reviewer gSQC,
>
> We would like to express our sincere gratitude once again for your positive feedback, insightful comments, and constructive suggestions. Your guidance has been invaluable in helping us improve the quality of our work! With sincere dedication, we have carefully addressed your remaining concerns on **MInference comparison**, **key self-similarity**, **overhead in no-offloading setup**,  **dynamic vs. fixed rules**, and **related work**, as detailed in our preceding responses—in response to your invaluable comments.
>
> We are writing to gently remind you that **the author-reviewer discussion period will end in less than 12 hours**. We eagerly await your feedback **to understand if our responses have adequately addressed your concerns. If so, we would deeply appreciate it if you could raise your score**. If not, we are eager to address any additional queries you might have, which will enable us to further enhance our work.
>
> Once again, thank you for your kind support and constructive suggestions!
>
> Best,
>
> Authors

---

### Official Review · Reviewer_spBc · 2025-07-01

**Clarity:** 3
**Significance:** 4
**Originality:** 3
**Rating:** 5
**Confidence:** 4

**Summary:**

The paper presents AttentionPredictor, an innovative approach for KV cache compression in large language models (LLMs) by predicting next-token attention scores during the decoding phase. The method employs a lightweight convolutional model to identify and retain only critical tokens, significantly reducing memory usage while maintaining model performance. Additionally, the authors propose a cross-token prefetching framework to hide the computational overhead of attention score prediction. The approach achieves impressive results, including 13$\times$ KV cache compression and 5.6$\times$ speedup while maintaining comparable performance to full cache inference.

**Questions:**

See weaknesses.

**Ethical Concerns:**

["NO or VERY MINOR ethics concerns only"]

**Final Justification:**

The authors have adequately addressed my concerns about the terminology, the architecture and the overhead of the proposed method. Therefore, I will keep my score.

**Limitations:**

See weaknesses.

**Quality:**

3

**Strengths And Weaknesses:**

Strengths:
1) The idea of learning-based attention predictor is novel. It also surprises me that the learned predictor generalizes well on unseen datasets.
2) The unified convolutional model is remarkably lightweight (only 21KB) compared to alternatives like SeerAttention (101MB), making it practical for deployment.
3) Comprehensive evaluations across multiple benchmarks demonstrate consistent performance improvements over state-of-the-art methods.
4) The paper is well-motivated and clearly written.

Weaknesses:
1) It is a little confusing that since we get the indices of critical tokens from the attention predictor (see Algorithm 1), does it mean we need to keep all KV caches? If so, it is inappropriate to claim it is a method for KV cache compression.
2) While the training process is described in Appendix B.1, more details about the CNN architecture (number of layers, kernel sizes, etc.) would be helpful.
3) The computational overhead of the prediction model itself could be better quantified, especially since it runs at every decoding step.

---

> ### Author Rebuttal · Authors · 2025-07-31
>
> We thank the reviewer for the insightful and valuable comments. We are encouraged by your positive feedback on the novelty, generalization, and lightweight nature of our approach. We respond to each comment as follows and sincerely hope that our rebuttal could properly address your concerns.
>
> ### W1: Clarification on the term "KV Cache Compression".
> > It is a little confusing that since we get the indices of critical tokens from the attention predictor (see Algorithm 1), does it mean we need to keep all KV caches? If so, it is inappropriate to claim it is a method for KV cache compression.
>
> Thank you for this insightful question. **You are correct** that **we maintain the full KV cache in CPU memory**. Our method achieves "compression" by **involving only a subset of the KV cache in the attention computation**, effectively reducing the active GPU memory footprint, which is the primary bottleneck during inference.
>
> Terms such as "**KV cache retrieval**" or **"attention sparsity**" may describe our method more precisely. We will revise the terminology in the final version of the paper to ensure greater clarity.
>
> ### W2: Details of the CNN Architecture.
> > While the training process is described in Appendix B.1, more details about the CNN architecture (number of layers, kernel sizes, etc.) would be helpful.
>
> Thank you for the suggestion. To provide a clearer and more quantifiable overview of our CNN architecture, we will add the following detailed breakdown to the paper's appendix.  **The architecture is in Table 1**. This design choice makes the model lightweight and adaptable to varying sequence lengths.
>
> ***Table 1**: AttentionPredictor Structure.*
> | Layer No. | Layer Type | Details |
> | :--- | :--- | :--- |
> | 1 | 2D Convolution | **Kernel**: (3, 3), **Padding**: 1, **Channels**: 1 → 16 |
> | 2 | ReLU Activation | - |
> | 3 | 2D Convolution | **Kernel**: (3, 3), **Padding**: 1, **Channels**: 16 → 32 |
> | 4 | ReLU Activation | - |
> | 5 | Adaptive Avg. Pooling | Collapses the temporal history dimension to size 1 |
> | 6 | 1D Convolution | **Kernel**: 1, **Channels**: 32 → 1 |
>
>
>
> ### W3: Computational Overhead of the AttentionPredictor.
> > The computational overhead of the prediction model itself could be better quantified, especially since it runs at every decoding step.
>
> Thank you for raising this important point. To clarify, the predictor **does not run at every decoding step**. To balance performance and overhead, we invoke the predictor periodically. In our experiments, we **update the critical token set only every two steps**, which effectively amortizes the prediction cost.
>
> We have quantified **the overhead of our predictor model**. As shown in **Table 2**, both the latency and memory footprint are minimal, ensuring that the prediction process does not become a bottleneck. We will include this analysis in the final paper.
>
> ***Table 2**: The computational overhead of the prediction model.*
> | Sequence Length | Prediction Latency (ms) | Peak Memory Usage (MB) |
> |---|---|---|
> | 4K | 0.28 | 97 |
> | 8K | 0.47 | 194 |
> | 16K | 0.91 | 388 |
> | 32K | 2.22 | 776 |
>
> Additionally, **Table 3** shows a detailed breakdown of our cache prefetch system.
> Our cross-token prefetching framework is illustrated in the timeline in **Figure 10 (Appendix B.1)**.
>
> ***Table 3:** Computation breakdown at 1/16 budget (in ms/token).*
> Context Length|Attention Prediction|Cache Transfer|Sync/Wait for next token|Total Overhead
> -|-|-|-|-
> 4K|1.4|2.3|73.7|77.3
> 8K|2.3|3.9|72.3|78.6
> 16K|4.6|7.6|67.3|79.6
> 32K|9.1|13.2|57.7|80.0
>
> The "Attention Prediction" latency in Table 3 is greater than the values in Table 2 because it encompasses not only the prediction model inference but also additional matrix operations essential for the prefetching process.
>
> Note that since we perform attention prediction once every two tokens, the average per-token inference latency is further reduced.

---

> > ### Comment · Reviewer_spBc · 2025-08-05
> > **Response to the Rebuttal**
> >
> > Thank you for the detailed clarifications. I have no further questions. Therefore, I will keep my rating of Accept.

---

### Note · Authors · 2025-08-12

Dear Area Chair,

Thank you for your guidance and support during the review process. We summarize our responses to the reviewers' concerns below.

### Summary of our responses
We are encouraged by the positive feedback, with reviewers highlighting our work is "**novel**" (spBc), "**interesting**" (Y2CJ), "**well-motivated**" (spBc), "**clearly written**" (spBc), and shows "**consistent performance improvements**" (gSQC, spBc) with a strong "**theoretical basis**" (Y2CJ, 8ZdG). Furthermore, We have addressed all remaining concerns with additional experiments.

**Reviewer spBc** has replied that "I will **keep my rating of Accept**."

**Reviewer 8ZdG** has replied that "I still **remain positive about the submission**."

**Reviewer Y2CJ** has replied that "After reading the comments, I have decided to **maintain my original score**." We would like to note that the reviewer did not provide additional comments or questions following our further response. We believe our replies and new experiments have fully addressed their concerns, summarized below:
- **Seasonal pattern mechanism**: Clarified the mechanism with practical examples and theoretical analysis.

- **Applicability to short contexts**: Confirmed our method's effectiveness on short-text scenarios via new experiments on two benchmarks (MMLU CoT, GPQA).

**Reviewer gSQC** has replied that "I'm looking forward to hearing your answers, I plan to **increase my score**". We provided a detailed response to their follow-up questions, and **have not received further comments**. We believe our replies and new experiments have fully addressed their concerns, summarized below:
- **Fairness of comparison**: We have clarified the fairness of our evaluation setup and further demonstrated the superior performance of our method through new experiments on a more challenging, dynamic baseline.
- **Computational overhead**: We have provided a detailed latency breakdown showing the overhead of our predictor is minimal (<1% of inference time).
- **Lack of evidence for Key stability**: We presented new empirical results that quantitatively confirm the high self-similarity of Keys, strengthening the theoretical foundation of our work.
- **Extra Related Work**: We have clarified our novelty as the first method based on direct attention pattern prediction, and better distinguished its advantages over other learning-based approaches.

Once again, thank you very much for your time and efforts.

Best regards,

Authors

---

### Decision · Program_Chairs · 2025-09-17

**Decision:**

Accept (poster)

**Comment:**

***Summary***

A great deal of research is currently focused on optimizing attention by reducing the size of the KV-cache, often by pruning unnecessary tokens.  In this paper, the authors introduce AttentionPredictor, which uses a small, lightweight, convolutional model to predict what portions of the cache will be necessary for the next decoding step.  The entire KV-cache must still be held in main memory on CPU, but only the predicted portions are prefetched and transferred to GPU, and the model only attends over the predicted portions, thus speeding up inference.

The authors do a qualitive analysis of attention patterns, and identify several temporal patterns that can be seen in attention, which they label "re-access", "sequential", and "seasonal".  The focus on temporal patterns distinguishes this work from prior work.  The authors also train a small 2D convolutional NN to predict these patterns, and use it to prefetch tokens.

The authors compare their technique against a number of other KV-cache compression baselines, on a variety of tasks, and show that their method generally outperforms competitors.  They also show large latency improvements against a FlashAttention baseline at long context lengths.

***Meta-review***

Reviewer opinions on this paper were somewhat mixed and in the borderline range, with ratings of (5, 4, 4, 3).   As AC, I have gone over both the paper and the discussion, and I believe these ratings are too low.  This is an excellent paper, and should be clearly accepted.

I am basing my decision on two factors.  First, the author rebuttal during the discussion period was very strong; the authors provided detailed and clear responses to all of the reviewer questions, and I do not believe that the current ratings reflect the strength of the rebuttal.  Second, I honestly think the ratings were too low to start with; I give my own opinions on the paper at the end of this review.

The most detailed review by far was done by reviewer **gSQC (rating 4)**, who asked a number of good technical questions (thank you!). The authors gave a very long and detailed response, and there was further discussion afterwards.  In their final justification, reviewer gSQC writes "The authors clarified all of my concerns in the rebuttal... I am raising my score to 4.  I'm keeping it at 4 because the work targets the KV off-loading setup which is more niche, therefore less significant."  As AC, I appreciate the high-quality review from gSQC, but I disagree with their assessment of significance.  If there are no major technical issues, then I believe this work is significant enough to get a rating of 5; KV off-loading is a commonly used technique.

The remaining reviews were all rather cursory, but the authors nonetheless wrote detailed responses to each reviewer concern.  Some examples are:

**Reviewer spBc (rating 5)**:
* The paper was "well-motivated and clearly written."  (As AC, I wholeheartedly agree.)
* "more details about the CNN architecture (number of layers, kernel sizes, etc.) would be helpful."  (The authors provided additional details, as requested.)

**Reviewer 8ZdG (rating 4)**:
* "Sections C.5 and 4.6 provide standard deviations but do not specify the error computation method."  (The authors provided the appropriate equations and justification.)
* "The cross-token prefetching framework claims to hide latency (Figure 5), but lacks a detailed breakdown of overhead components (e.g., prediction time, data transfer, synchronization)."  (The authors provided a table listing overheads.)
*  "no deployment data for edge devices is provided"  (Author response: "Our work primarily targets large-scale, cloud-based LLM inference systems.")

Reviewer 8ZdG's response was: "The rebuttal... addresses most of my concerns raised in the initial review."

**Y2CJ (rating 3)**  was unconvincing to me, and the reviewer did not adequately respond to the author rebuttal.
* "The writing remains a large improvement space."  (As AC, I strongly disagree with this statement.)
* "The evaluation are conducted on narrow tasks, a simple needle-in-a-haystack task and a longbench task."  (Factually wrong, and corrected by authors in rebutttal.)
* "The calibration step M is 5, a small number. AKA, every 5 steps the proposed method needs to 1) compute attention scores on the whole sequence; 2) fetch KV cache of the whole sequence."  (The authors responded with a detailed explanation, and an experiment ablation with various different values for M.)

***AC opinion***

This paper has a number of strengths.

* The paper is very clear and extremely well-written, among the best in my batch.  It provides qualitative assessments, which give further insight into attention in general, along with solid quantitative analysis and good diagrams.
* The literature review is fantastic.
* The problem is important, and the solution seems novel to me.
* The authors discuss various subtleties that weaker papers often overlook, e.g. line 240, "distributional error calibration."
* The implementation of their technique is solid, as can be seen from the author responses during rebuttal.
* The experiments are comprehensive and convincing.